Journal of Data-centric Machine Learning Research (2024)          Submitted 7/24; Revised 9/24; Published 9/24

# When is Off-Policy Evaluation (Reward Modeling) Useful in Contextual Bandits? A Data-Centric Perspective

**Hao Sun**\*, **Alex Chan**\*, **Nabeel Seedat**, **Alihan Huyuk**, **Mihaela van der Schaar**
{HS789,AJC340,NS741,AH2075,MV472}@CAM.AC.UK
*Department of Applied Mathematics and Theoretical Physics*
*University of Cambridge*
*Cambridge, UK*

**Reviewed on OpenReview:** *https://openreview.net/forum?id=wg5y4AK6l7*

**Editor:** Omar Rivasplata

## Abstract

Evaluating the value of a hypothetical target policy with only a logged dataset is important but challenging. On the one hand, it brings opportunities for safe policy improvement under high-stakes scenarios like clinical guidelines. On the other hand, such opportunities raise a need for precise off-policy evaluation (OPE). While previous work on OPE focused on improving the algorithm in value estimation, in this work, we emphasize the importance of the offline dataset, hence putting forward a data-centric framework for *evaluating OPE problems*. We propose DataCOPE, a data-centric framework for *evaluating OPE* in the logged contextual bandit setting, that answers the questions of whether and to what extent we can evaluate a target policy given a dataset. DataCOPE (1) forecasts the overall performance of OPE algorithms without access to the environment, which is especially useful before real-world deployment where *evaluating OPE is impossible*; (2) identifies the sub-group in the dataset where OPE can be inaccurate; (3) permits evaluations of datasets or data-collection strategies for OPE problems. Our empirical analysis of DataCOPE in the logged contextual bandit settings using healthcare datasets confirms its ability to evaluate both machine-learning and human expert policies like clinical guidelines. Finally, we apply DataCOPE to the task of reward modeling in Large Language Model alignment to demonstrate its scalability in real-world applications.

**Keywords:** Data-Centric AI, Off-Policy Evaluation, Contextual Bandits, Reward Modeling

## 1 Introduction

Introducing novel policies and guidelines in high-stakes settings such as healthcare and criminal justice comes with great potential harm, and should be backed by appropriate data and evidence before enacting the policy or guideline (Woolf et al., 1999; Suresh and Guttag, 2021). The challenge is that it can be hard to actually know when your data is sufficient, and without the ability to run a policy to see what *actually* happens, we must resort to estimating its effect given a logged *offline dataset* of previously seen observations and actions. Generally, this is known as **off-policy evaluation** (OPE) (Precup et al., 2000; Beygelzimer and Langford, 2009; Dudík et al., 2011), using data collected previously under one policy to predict the performance of *another* policy. While this terminology emerged

in the reinforcement learning (RL) literature, it is rooted in a causal problem and is often considered under the guise of treatment effect estimation when intervention occurs only in one time step (Powers et al., 2018). The causal nature highlights why the offline RL problem is hard, and there is a limit to what can be said about a policy from *observational* data alone (Pearl, 2009).

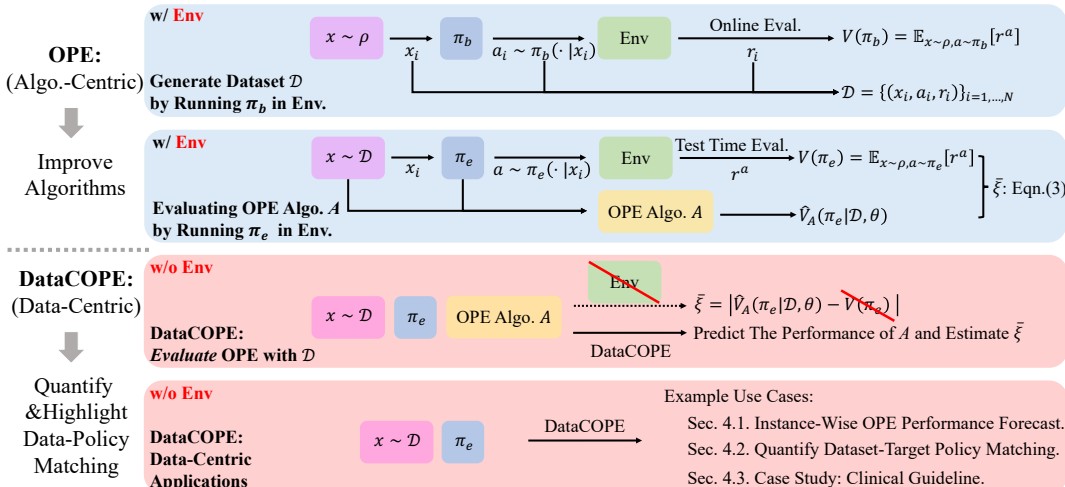

Figure 1: *Road map of DataCOPE.* To highlight the difference between DataCOPE and classical OPE literature: the objective of DataCOPE is to evaluate whether OPE problems are well-defined, while OPE focus on improving estimators. **1-st row:** illustrates the offline dataset collection process. In the context of healthcare, it corresponds to treatment records abiding by an existing guideline. **2-nd row: (Sec.2)** The collected dataset $\mathcal{D}$ is then used for OPE. For an OPE algorithm, Equation (3) calculates the test-time residual (error) between the value estimation result from an algorithm and the true value. **3-rd row: (Sec.3)** DataCOPE can serve as a proxy for the evaluation residual. It can work in test-time as an OPE performance indicator without access to the true value $V(\pi_e)$ and the environment. **4-th row: (Sec.4)** DataCOPE can be applied to various use cases, which is demonstrated with extensive empirical studies. Notions are explained in Sec.2.

OPE has been extensively studied both methodologically and empirically (Strehl et al., 2010; Jiang and Li, 2016). However, this has mostly been from the perspective of improving estimators. As such, we challenge the underlying assumption that the dataset is suitable for all potential OPE methods, a neglected focus so far. In this work, we take a *data-centric* approach, considering the problem: **Given a *dataset*, how accurately can OPE algorithms evaluate *a specific target policy*?**

Throughout the paper, we will return to the concrete example of introducing a new clinical guideline (meaning a policy), such as the Model for End-stage Liver Disease (MELD) scoring in liver transplant allocation (Habib et al., 2006). These are highly impactful, committee-made decisions, for which we really must know the answers to the question of whether it will be effective and if so, who will it actually benefit?

**What is needed from the community?** We require a method that can *evaluate whether an OPE method will be reliable* and *identify for what contexts/policies an evaluation is uncertain.* For clarity, consider the following desiderata: **(D1)** Method-Agnostic (Data-

Centric): it should be robust to change of underlying OPE methods, being able to capture the intrinsic difficulty of OPE problems regardless of the methods being selected. **(D2)** Individualized Evaluation: for hard-to-evaluate policies, it should be able to identify the reason (examples) that causes the difficulty in evaluation. Specifically, it will be useful if hard-to-evaluate examples can be discovered. **(D3)** Assess Data-Policy Matching: It should be able to identify which dataset, or data collection strategy, is more appropriate in evaluating a certain target policy.

Fulfilling D1-D3, in this work we propose Data-Centric Off-Policy Evaluation (DataCOPE), a framework that evaluates the inherent difficulty of OPE problems and is able to predict the general performance of OPE algorithms by decomposing the estimation uncertainty in the problem. DataCOPE detects dataset-target policy mismatch and thus compares data collection strategies for more accurate OPE without an environment. Figure 1 illustrates how this paper develops. Contributions of our work are threefold:

1. Methodologically, our research diverges from previous algorithm-centric studies on OPE, which have primarily concentrated on developing algorithms. Instead, our investigation places a significant emphasis on the crucial role of data, particularly with regard to the target policy, in OPE problems. Thus, this work is the first to bring ideas of data-centric AI from supervised learning to address the unique issues pertinent to OPE.
2. Practically, we introduce DataCOPE as an evaluation proxy for OPE problems. Traditional evaluation of OPE algorithms requires access to the true target policy value or live environment, but DataCOPE can serve as a proxy to predict whether OPE algorithms perform well in the absence of an environment.
3. Empirically, we demonstrate that DataCOPE (1) serves as an effective evaluation proxy for OPE, (2) provides a detailed performance prediction on instance-wise value estimation, and (3) can be applied to real-world datasets, such as evaluating clinical guidelines.

## 2 Preliminaries

We focus on a **logged contextual bandit** setting (Joachims et al., 2018). In particular, we consider **contexts** $x \in \mathcal{X}$, **actions** $a \in \mathcal{A} := \{1, 2, ..., k\}$, and **rewards** $r^a \in \mathbb{R}^+$ generated by the stochastic reward generation process taking action $a$ given context $x$. In this environment, a learner can act according to a **policy** $\pi \in \Pi := \Delta(\mathcal{A})^{\mathcal{X}}$, the **value** of which is given by:

$$V(\pi) = \mathbb{E}_{x \sim p(X), a \sim \pi(x)} \left[ r^a \right], \tag{1}$$

the expected reward obtained by executing the given policy. A **contextual bandit** problem then involves finding the solution to:

$$\pi^* = \arg \max_{\pi \in \Pi} V(\pi), \tag{2}$$

considered the **optimal policy** $\pi^*$. Typically this is solved via repeated interaction with the environment, which is not available when we move to the **logged** setting. In this case, we are unable to interact with the environment but alternatively have access to a **dataset** $\mathcal{D} = \{(x_i, a_i, r_i)\}_{i=1}^N$ of context, action, reward tuples. These have been generated via a **behavior policy** $\pi_b \in \Pi$ that we *do not* observe but has previously interacted with the environment as follows:

1. An examples $x_i \sim p(X)$ is drawn.
2. The behavior policy $\pi_b$ selects action $a_i \sim \pi_b(x_i)$.
3. A stochastic reward $r^a$ is observed.
4. $(x_i, a_i, r_i)$ is added to $\mathcal{D}$.

This makes the optimization task in (2) difficult as in the dataset $a_i \sim \pi_b$, which will introduce significant bias if we attempt to Monte Carlo estimate the expectation directly using our given samples. Thus, a large part of the logged contextual bandit problem revolves around the accurate estimation of (1), a task referred to as **Off-Policy Evaluation** (OPE), as we wish to *evaluate* a policy using data collected *off-policy* (i.e., with a behavioral policy).

Many algorithms have been proposed for such estimation, the learning objective of which is normally to minimize the mean-square error $\bar{\xi}$ (**OPE residual**) between real and estimated values. Considering a value estimator $\hat{V}_A(\pi|\mathcal{D}, \theta)$, built with dataset $\mathcal{D}$ and parameterised by $\theta$, the learning objective of the algorithm $A$ is to minimise:

$$\bar{\xi}_A := \text{MSE}(\hat{V}_A) = \mathbb{E}\left[\left(V(\pi) - \hat{V}_A(\pi|\mathcal{D}, \theta)\right)^2\right]. \tag{3}$$

**Plug-in Estimation of the Value Function.** An important sub-class of OPE algorithms, based on the **Direct Method** (DM) (Beygelzimer and Langford, 2009) revolves around constructing a directly parameterized estimator of the reward, a function $\hat{q}_\theta$ taking contexts and actions and returning the predicted reward. This is typically learned using supervised learning based on the dataset $\mathcal{D}$:

$$\hat{q}_\theta = \arg\min_{\hat{q}_\theta} \mathbb{E}_{(x,a,r)\sim\mathcal{D}}\left[(r - \hat{q}_\theta(x,a))^2\right]. \tag{4}$$

Armed with $\hat{q}_\theta$, DM estimates the predicted reward of taking actions according to the policy and plugs them into the value function calculation, producing an estimator:

$$\hat{V}_{\text{DM}}(\pi|\mathcal{D}, \theta) = \mathbb{E}_{x\sim\mathcal{D}, a\sim\pi}\left[\hat{q}_\theta(x,a)\right]. \tag{5}$$

This class of methods is by far one of the most popular (Saito et al., 2020; Fu et al., 2021) (and the one we shall build our method around).

**Data Characterization.** Previous data-centric works like Data-IQ (Seedat et al., 2022) and Data Maps (Swayamdipta et al., 2020) evaluate the data in classification settings with the goal of characterizing examples in a dataset into easy, hard and ambiguous based on analyzing the training dynamics of individual examples (Seedat et al., 2024). Such methods assume access to the prediction probability for the ground-truth class to compute uncertainty measures and are largely focused on curating a high-quality training dataset (Seedat et al., 2023). However, the problem of OPE has three distinct differences: (1) we are focused on test time evaluation, where (2) the ground-truth label in value prediction is not available, therefore, the prediction confidence used in prior works is unavailable and (3) these methods are unable to tackle regression tasks without a softmax probability, which are always needed in OPE. Extended discussions on related work are elaborated in Appendix A.

## 3 When is OPE Useful?

Given a dataset, target policies are not created equal for OPE. Consider an offline dataset generated by some behavior policy $\pi_b$, then intuitively the more similar a target policy $\pi_e$ is

to the behavior policy the easier it should be to evaluate the performance of $\pi_e$ using the dataset generated by $\pi_b$. When the decisions made by $\pi_e$ and $\pi_b$ are similar, outcomes of the actions from $\pi_e$ should be well represented in the dataset, while the unsupported actions' values will be challenging to predict. However, without explicit access to the behavior policy $\pi_b$, measuring its distance to the target policy is a highly non-trivial task.

### 3.1 Inherent Difficulty of OPE: A Data-Centric Perspective

A critical objective of this work is to provide a forecast of how accurately an OPE problem can be solved from a data-centric perspective: we emphasize the importance of *data*, rather than the algorithms. Such a perspective permits a hierarchical analysis of the problem:

First and foremost, we want to provide an overall description of the *inherent difficulty* of the OPE problem. Given the offline dataset $\mathcal{D}$, and target policy $\pi_e$, we formally write the difficulty of the OPE problem as $\mathcal{H}$. Intuitively, it describes the expected OPE residual over different instantiations of algorithms given a behavior dataset and a target policy:

$$\mathcal{H}(\mathcal{D}, \pi_e) \propto \mathbb{E}_\theta \text{MSE}(\hat{V}_A(\pi_e | \mathcal{D}, \theta)), \quad \forall A \tag{6}$$

where $A$ denotes an arbitrary OPE algorithm and $\theta$ is its stochastic instantiation, considering different initialization and optimization processes. Then, an OPE problem evaluation method should permit a case-by-case analysis of the OPE problem, and give fine-grained explanations of the difficulties identified above. In doing so, it should identify the sources of OPE difficulty. While in some cases the difficulty originates from inherent stochasticity, in some other cases it requires more diverse samples to match the target policy for an accurate OPE.

### 3.2 Distributional Direct Method for Uncertainty Decomposition

A central part of our idea for quantifying the difficulty of OPE is to quantify decomposed uncertainty and use that to build a model that is capable of predicting the OPE residual. As is common in data-centric approaches, to satisfy the previous desiderata we propose to separate the aleatoric (data) uncertainty and epistemic (model) uncertainty parts of the prediction (Seedat et al., 2022), which in our case is the value estimator. This has many benefits, especially for comparing datasets for OPE, since the only relevant part here is the epistemic (model) uncertainty.

This section is organized as follows: first, we propose our tailored DM that has a distributional reward estimator; we then introduce the uncertainty decomposition used for breaking down this estimator; finally, we present a practical method for predicting OPE difficulty using these components.

**DM with Distributional Reward Estimators** To achieve such a separation, instead of using a regular regression model that only learns an expected value, we leverage a probabilistic network to *capture the distributional information* in value estimation. When the reward model is binary (i.e., only success with +1 or failure with 0), a network with a standard softmax output can be considered to output the logits of a Bernoulli distribution. For continuous reward models, normal regression models are insufficient and so we adopt the DM with a mixture density network (MDN) (Bishop, 1994) as a reward estimator. Specifically, an MDN predicts a set of parameters used in a mixture of Gaussian predictive distribution so as to maximize the log-likelihood of observed data. The likelihood with $K$ Gaussian is

$$\mathcal{L} = \sum_{i=1}^{i=\mathbb{N}} \mathcal{L}(r_i|x_i, a_i, \theta) = \sum_{i=1}^{i=\mathbb{N}} \sum_{k=1}^{K} w_k(x_i, a_i, \theta) \times \phi(r_i|\mu_k(x_i, a_i, \theta), \sigma_k(x_i, a_i, \theta)), \quad (7)$$

where $\theta$ denotes the parameters of networks with three branches of outputs $\{w_k, \mu_k, \sigma_k\}_{k=1,...,K}$, $\phi$ is the probability density function of normal distribution , and $\sum_{i=1}^{K} w_i = 1$ is the normalized weight. In this case, the reward prediction, given context $x$ and action $a$, is a random variable, denoted as $\hat{R}_x^a$.

**Uncertainty Decomposition**  As we are interested in the predictive random variable $\hat{R}_x^a$, we model the uncertainty with value predictive variance: $v(x, a) = \mathbb{V}_{\hat{R}^a|X=x, A=a}(\hat{R}^a|X = x, A = a)$. According to the law of total variance, we have:

$$v(x, a) = \mathbb{V}_{\Theta}\left[\mathbb{E}_{\hat{R}^a|X=x, A=a}\left(\hat{R}^a|X = x, A = a, \Theta\right)\right] + \mathbb{E}_{\Theta}\left[\mathbb{V}_{\hat{R}^a|X=x, A=a}\left(\hat{R}^a|X = x, A = a, \Theta\right)\right],$$
$$(8)$$

where $\Theta$ is a random variable that has an empirical distribution over the set of parameters with different instantiations, i.e., ensemble.

In this way, the overall uncertainty is split into two components: the *epistemic uncertainty* component $v_{\mathrm{ep}} = \mathbb{V}_{\Theta}\left[\mathbb{E}_{\hat{R}^a|X=x, A=a}\left(\hat{R}^a|X = x, A = a, \Theta\right)\right]$, and the *aleatoric uncertainty* component $v_{\mathrm{al}} = \mathbb{E}_{\Theta}\left[\mathbb{V}_{\hat{R}^a|X=x, A=a}\left(\hat{R}^a|X = x, A = a, \Theta\right)\right]$. Regarding the epistemic component, represented by $v_{\mathrm{ep}}$, the variance emerges from model fluctuations due to insufficient training data. Reducing this variance can be achieved by acquiring more data. In contrast, the aleatoric component, $v_{\mathrm{al}}$, is rooted in the intrinsic uncertainty within the data that obstructs precise predictions in certain contexts. Here, augmenting the dataset with more informative features, rather than merely increasing data volume, would be more helpful.

**Residual Prediction through Uncertainty**  Now with the decomposition of uncertainty, we are able to compare the evaluation confidence of target policies: naturally, policies that induce $(x, a)$ pairs with higher epistemic and aleatoric uncertainties are harder to evaluate. While the former can be alleviated by collecting more examples in the training data, the latter is an inherent problem of the task. In the following, we further introduce a residual predictor that can be built on a separate validation set to quantify how good are those uncertainty components in predicting the OPE residuals. We note that the technical contribution of DataCOPE does not rely on such a step, and we will demonstrate this through later experiments. The difficulty of OPE problems is correlated with our decomposed uncertainties. Such a decomposition, without this predictor, is useful in that it permits a comparison among datasets to answer the question of "*which dataset is most appropriate in evaluating a target policy*" (Appendix C); and it permits a hindsight interpretation of clinical guideline deployment and vulnerable group identification. (Section 4.2)

To quantitatively estimate the accuracy of OPE algorithms with the help of training data: Given the uncertainty decomposition of $(x, a)$ pairs, we are able to build a linear regression model $h$ that takes $v_{al}, v_{ep}$ as the inputs and outputs the OPE residual. We call this model $h$ the hardness predictor, and fit it with a group of held-out training data from $\mathcal{D}$ where true residual can be obtained without the environment:

$$h^* = \arg\min_{h} \mathbb{E}_{(x,a) \sim \mathcal{D}}(\bar{\xi} - h(v_{\mathrm{ep}}(x, a), v_{\mathrm{al}}(x, a)))^2, \quad (9)$$

where $\bar{\xi}$ is defined in Eqn.(3). When evaluating actions generated by a target policy, such a model $h$ is able to work as a hardness indicator function and predicts OPE residuals of the

given target policy. We provide a pseudo-code of DataCOPE in Appendix D, and elaborate on more implementation details in Appendix E.

The pseudocode of DataCOPE is provided in Algorithm 1.

---

**Algorithm 1** DataCOPE

---

**Input** Logged Dataset $D = \{x_i, a_i, r_i\}$, target policy $\pi_e$.
**Output** $h$ that Forecasts OPE Residual
\# 1. Build Value Estimator
    Optimize $\hat{q}_\theta$ with Equation (7) for non-binary reward estimation and classifiers with logits outputs otherwise.
\# 2. Uncertainty Decomposition
   Decompose the uncertainties in estimating $\hat{q}_\theta$ with Equation (8).
\# 3. (Optional) Calibration
   Quantify OPE residual according to Equation (9).
**Return**
    Difficulty in estimating different examples

---

## 4 Experiments

Recall the three desiderata (and hence goals of our method) that we established in the Introduction: we want to be able to 1) identify an instance-wise difficulty in evaluation, 2) be able to compare datasets for evaluating a certain policy by comparing coverage, and 3) robustly evaluate arbitrary OPE algorithms. In experiments, we evaluate DataCOPE's ability to fulfill them point-by-point in Section 4.1, Appendix B and  C limited by the space, alongside ablations that consider how our ability is affected by factors such as policy complexity and bias, as well as data coverage. Section 4.2 further verifies those abilities on a real-world clinical dataset. We demonstrate how DataCOPE can be applied to evaluate clinical guidelines.

**Synthetic Dataset Generation** Following standard procedures in the OPE literature (Chu et al., 2011; Li et al., 2012; Agrawal and Goyal, 2013), we adapt supervised learning datasets into example logged bandit datasets where we know the true underlying generative process. In particular, we use two medical tabular datasets, namely `Breast Cancer` and `Diabetes` (Dua and Graff, 2017), representing both classification and regression tasks respectively, to validate that DataCOPE is effective and powerful. Further validation on other common UCI datasets is provided in Appendix F to verify the generalization of DataCOPE.

We use linear models for the behavior policy $\pi_b$, which is trained on the dataset examples $\{(x_i, y_i)\}_{i=1}^N$. Given context $x_i$ with corresponding labels $y_i$ we learn a policy to generate actions by minimizing the expected negative log-likelihood $\mathbb{E}_{x,y}\big[\mathrm{NLL}(\pi_b(x_i), y_i)\big]$, under a predictive Gaussian/Bernoulli distribution for regression/classification respectively.

We then evaluate $(x_i, a_i)$ with the help of ground-truth labels $y_i$. For classification tasks, the reward of action $a$ is given as $r_i^a = \mathbf{1}(a_i = y_i)$, where $\mathbf{1}$ is the indicator function; while for regression tasks, the reward of action $a_i$ is given by the coefficient of determination of the prediction. i.e., $r_i^a = 1 - \frac{u}{v}$, where $u$ is the residual sum of squares $u = ||y - a||_2^2$ and $v$ is the total sum of squares $v = ||y - \bar{y}||_2^2$, $\bar{y}$ denotes the mean of $y$. In this way, we can generate $\{(x_i, a_i, r_i^a)\}_{i=1}^N$ tuples as our dataset containing $N$ examples.

Table 1: *DataCOPE is able to predict the instance-wise evaluation error of various OPE algorithms.* The numbers in the first rows of every (4-row) block are correlations between DataCOPE's two uncertainty components and OPE residuals under different settings. The other three rows are the ablation studies (i.e., correlation drop) by removing the aleatoric uncertainty component, removing the epistemic uncertainty component, and not doing uncertainty decomposition. ***Higher is better.***

| Dataset | Ablations | DM | DR | RM | SNIPW | IPWsk | SNDR |
|---|---|---|---|---|---|---|---|
| Breast Cancer | DataCOPE | 0.708 | 0.899 | 0.936 | 0.936 | 0.936 | 0.901 |
| | w/o $v_{\text{al}}$ | 0.572 $(\downarrow 0.136)$ | 0.539 $(\downarrow 0.360)$ | 0.486 $(\downarrow 0.450)$ | 0.486 $(\downarrow 0.450)$ | 0.486 $(\downarrow 0.450)$ | 0.537 $(\downarrow 0.364)$ |
| | w/o $v_{\text{ep}}$ | 0.579 $(\downarrow 0.129)$ | 0.855 $(\downarrow 0.044)$ | 0.914 $(\downarrow 0.022)$ | 0.914 $(\downarrow 0.022)$ | 0.914 $(\downarrow 0.022)$ | 0.858 $(\downarrow 0.043)$ |
| | w/o Decomposition | 0.579 $(\downarrow 0.129)$ | 0.855 $(\downarrow 0.044)$ | 0.914 $(\downarrow 0.022)$ | 0.914 $(\downarrow 0.022)$ | 0.914 $(\downarrow 0.022)$ | 0.858 $(\downarrow 0.043)$ |
| Diabetes | DataCOPE | 0.743 | 0.743 | 0.745 | 0.745 | 0.744 | 0.744 |
| | w/o $v_{\text{al}}$ | 0.728 $(\downarrow 0.015)$ | 0.729 $(\downarrow 0.014)$ | 0.732 $(\downarrow 0.013)$ | 0.732 $(\downarrow 0.013)$ | 0.731 $(\downarrow 0.013)$ | 0.731 $(\downarrow 0.013)$ |
| | w/o $v_{\text{ep}}$ | 0.386 $(\downarrow 0.357)$ | 0.378 $(\downarrow 0.365)$ | 0.374 $(\downarrow 0.371)$ | 0.374 $(\downarrow 0.371)$ | 0.374 $(\downarrow 0.370)$ | 0.374 $(\downarrow 0.370)$ |
| | w/o Decomposition | 0.386 $(\downarrow 0.357)$ | 0.379 $(\downarrow 0.364)$ | 0.375 $(\downarrow 0.370)$ | 0.375 $(\downarrow 0.370)$ | 0.374 $(\downarrow 0.370)$ | 0.375 $(\downarrow 0.369)$ |

**Target Policy**   We also generate target policies using neural network models trained on the *same* training data $\{(x_i, y_i)\}_{i=1}^N$, but with injected noise, that will depend on the particular experiment, as $\pi_e$. The ground-truth performance of policy $\pi_e$ is then given by $r_j^b = \mathbf{1}(b_j = y_j)$, where $b_j = \pi_e(x_j)$. Importantly, this is available to us and thus allows for the evaluation of the quality of the OPE. In particular, the off-policy evaluation problem is by definition estimating $\mathbb{E}[r_j^b]$, for $j = 1, ..., N$. As we have access to $y_j, j = 1, ..., N$, we can quantitatively evaluate our method and compare results with the ground-true policy values.

**General Experiment Settings**   We conducted all our experiments using 8 random seeds and reported the mean and standard deviation of the results. The observed performance differences with 8 trials were found to be statistically significant. For the ensemble-based uncertainty quantification, we determined that using 100 models was computationally feasible for tabular datasets in the healthcare domain. However, we also found that using 10 models produced satisfactory results. We note that DataCOPE is the first to tackle OPE problems by evaluating the problem themselves.

To maintain consistency with prior literature, we differentiated between the behavior policy and target policy in two distinct ways. The first approach involved injecting random noise into the labels or regression target during the generation of behavior policies, thereby ensuring that the behavior trajectories (i.e., dataset) did not completely conform to the target policy behavior. The second, slightly more advanced method involved biasing the policy through dataset sculpture, as described in Section 4.2. The aim of both approaches was to evaluate the performance of DataCOPE on various datasets with differing degrees of mismatch between the behavior and target policies.

## 4.1 Instance-Wise Difficulty Indication

In this section, we zoom in on the individuals who will be acted on by the policies and use DataCOPE to predict instance-wise OPE difficulty. We provide an analysis at the algorithm level, asking how well will an OPE method do *on average* in Appendix B.

**Experimental Setup**   For each dataset, we run different OPE algorithms and are able to calculate the value estimation residual (error) over every datum, individually comparing this to the output of DataCOPE, which has decomposed the uncertainties into aleatoric and

epistemic components. The correlation between the OPE residuals and the uncertainties is then calculated and reported, noting that a *high* correlation implies that the DataCOPE uncertainties have *predictive power* when it comes to determining if the OPE is accurate. We consider the following OPE algorithms: Direct Method (**DM**), the Doubly Robust Estimator (**DR**), Replay Method (**RM**), Self-Normalized Inverse Propensity Weighting (**SNIPW**), Inverse Propensity Weighting with Shrinkage (**IPWsk**), and Self-Normalized Doubly Robust (**SNDR**). Our implementation of OPE estimators is based on the open-sourced package available at `https://github.com/st-tech/zr-obp/tree/master/obp/ope`.

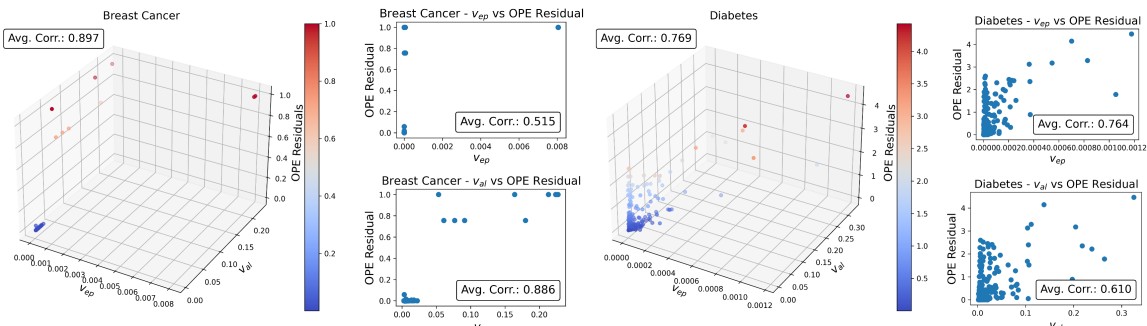

Figure 2: *DataCOPE decomposes the uncertainty and provides an instance-wise prediction of the estimation error.* In the 3-D plots, we use colors to highlight the averaged OPE residual values (also z-axis) and visualize their strong correlation with the two uncertainty components. In the 2-D plots, we visualize the correlation between each uncertainty component and the averaged OPE residual values. The results highlight the necessity of uncertainty decomposition as different components may dominate the prediction of OPE residuals. This high correlation holds for different algorithms. For detailed non-averaged results, please refer to Appendix H.

**Results** As shown in Table 1, DataCOPE effectively predicts the OPE residual, but we can see the effect on an individual level more clearly in the scatter plot of Figure 2 which qualitatively shows the relation between two uncertainties and the averaged OPE performance in terms of residual over 6 algorithms. Normally, the examples whose value can not be precisely estimated are located at the upper right in the scatter plots, meaning they either have high aleatoric or epistemic uncertainty.

Our ablation studies also serve to demonstrate how the decomposition is important: In `Diabetes`, the epistemic uncertainty is more important than the aleatoric component in predicting the OPE residual, indicating that such difficulty can potentially be alleviated by collecting more data. While in `Breast Cancer`, we find aleatoric uncertainty dominates the performance prediction. This is important to be aware of since the aleatoric dominance in `Breast Cancer` suggests there is limited opportunity in the future to be able to reduce this uncertainty by collecting additional data, which is not the case at all for `Diabetes`. In both cases, the decomposition is vitally important as it manifests the uncertainty components even at a different magnitude.

**Take-away:** *DataCOPE is able to predict instance-wise difficulty in OPE for both discrete and continuous tasks. The uncertainty decomposition step in DataCOPE isolates the uncertainty components' effect even when they are of different magnitudes.*

Table 2: DataCOPE strongly correlates with the OPE residual on the Organ Transplant dataset.

| DataCOPE | $0.712 \pm 0.069$ |
|---|---|
| w/o $v_{\mathrm{ep}}$ | $0.654 \pm 0.044$ |
| w/o $v_{\mathrm{al}}$ | $0.309 \pm 0.198$ |
| w/o Decomposition | $0.654 \pm 0.044$ |

### 4.2 Case Study: The Introduction of MELD

We apply DataCOPE to a real-world healthcare guideline and examine its potential to benefit high-stakes OPE problems, with a specific focus on evaluating organ transplant allocation policy.

**Background**  In the medical field, the official guidance on organ transplantation and which potential recipients are offered organs when they become available has evolved multiple times over the past few decades (Starzl et al., 1982; Adam et al., 2012; Hüyük et al., 2022; Chan et al., 2022). This setting can be seen as a contextual bandit problem where every arriving patient and organ feature instance is considered the context, while the policy makes an allocation decision as the action, with the corresponding survival time for the patient perceived as an internal reward. We examine data from the Organ Procurement & Transplant Network (OPTN) (Leppke et al., 2013), which includes information on patients registered for liver transplants from 1995 to 2020. More details on data processing can be found in Appendix E.

We specifically focus on the deployment of the prevailing allocation policy MELD (Bernardi et al., 2011) and study the OPE problem of MELD before its deployment time, 2002. We find DataCOPE:

1. has a strong correlation with the OPE residual;
2. predicts and explains when and why additional collected data improves OPE;
3. identifies vulnerable sub-groups of patients that are more likely to suffer from inaccurate OPE estimation, hence potentially having high risk in medical practice.

This is important to consider whether there actually was sufficient evidence at the point of deployment to justify if the guidance would be, or if there were some (sub-)groups who might be left worse off.

**Experiment Setting**  We consider the behavior policy $\pi_b$ to be that which was deployed before 2002. In order to approximate the process of selecting the most in-need patient, for each patient who receives an organ at some time point, we add 9 other contemporary patients not being allocated to the organ as reference examples. $\pi_b$ then identifies the selected patient out of the 10 patients with a discrete action, and the corresponding reward is given by the survival time of the selected patient after receiving the donated liver.

This is then used to estimate the value of the MELD policy, the $\pi_e$ in the context of this OPE problem, that is deployed after 2002. Similar to the training dataset, we collect test-time data using the transplant records between 2003 and 2005. As MELD was applied as the allocation policy during this period, we have the real value of $\pi_e$ — the average survival time of the patients who receive organs during this period. We apply the direct method as a demonstrative OPE solver in experiments considering it does not require a parameterized policy.

**Results: Correlation.** We calculate the Pearson R between uncertainty components calculated by DataCOPE and the OPE residual. Table 2 shows that DataCOPE is highly correlated with OPE performance. Our ablation studies highlight the importance of uncertainty decomposition.

**Results: Evaluating MELD Evaluation over time.** By utilizing DataCOPE, we are able to examine snapshots of off-policy datasets taken at different points in time, analyze the uncertainties, and compare them with the residuals resulting from OPE, as illustrated in Figure 3. 1. Before the year 2000, the Institute of Medicine allocation recommendations

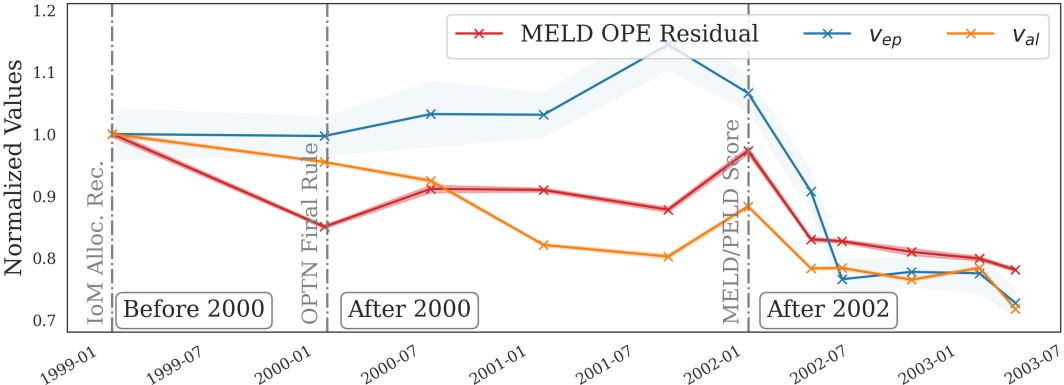

Figure 3: Evaluating MELD over time. DataCOPE can be applied to monitor the OPE performance without the real policy value.

(*IoM Alloc. Rec.*) were used as the behavior policy. Collecting more data during this period can improve the evaluation of the MELD policy. By comparing the results provided by DataCOPE before and at 2000, we can see that the epistemic uncertainty does not change significantly, while the aleatoric uncertainty decreases. This suggests that the OPE task should become easier, which is verified by the reduction in OPE residuals at 2000.

2. After the year 2000, the *OPTN Final Rule* was implemented as the allocation policy. This change in policy affected the pattern of collected data and subsequently, the performance of OPE for MELD. During the period when OPTN was in operation (2000-2002), the aleatoric component of uncertainty decreased while the epistemic component increased. This reminds us to be cautious when using data from this period to evaluate MELD. In fact, the OPE residual using data from this period increases.

3. After 2002, the *MELD* guideline was implemented as the allocation policy, replacing OPTN. As a result, the data collected since 2002 is unbiased. Both uncertainty components decreased significantly and so did the OPE residual when looking back in hindsight.

To summarize, our study on the Organ Transplant dataset shows that in order to achieve more accurate OPE results, efforts should be made to reduce the epistemic uncertainty as identified by DataCOPE.

**Results: Vulnerable Sub-Groups Identification with $v_{\text{ep}}$.** In this section, we use DataCOPE to investigate the sub-group with high epistemic uncertainty in OPE, as improving the performance of this sub-group is possible through collecting more data. We examine the

Table 3: Vulnerable sub-group identification. DataCOPE discovers the vulnerable group in OPE problem of MELD, and can be used to forecast the performance. **_Lower is Better_**

|  | Data | Sub-Group | Overall | Ratio |
|---|---|---|---|---|
| $v_{\mathrm{ep}}(10^{-5})$ | 2000 *(IoM)* | $2.261 \pm 0.124$ | $1.252 \pm 0.050$ | $\times 1.808$ |
|  | 2002 *(OPTN)* | $2.238 \pm 0.146$ | $1.358 \pm 0.054$ | $\times 1.648$ |
|  | 2003 *(MELD)* | $1.443 \pm 0.081$ | $1.151 \pm 0.048$ | $\times 1.254$ |
| $\bar{\xi}(10^{-2})$ | 2000 *(IoM)* | $8.869 \pm 0.116$ | $6.625 \pm 0.022$ | $\times 1.339$ |
|  | 2002 *(OPTN)* | $7.885 \pm 0.125$ | $6.838 \pm 0.023$ | $\times 1.153$ |
|  | 2003 *(MELD)* | $6.351 \pm 0.043$ | $6.098 \pm 0.022$ | $\times 1.041$ |

examples with the highest epistemic uncertainty in OPE, and found those selected patients, in general, have a larger *Weight Difference.*

To manifest how the performance of OPE evolves over such a sub-group, we compute the averaged OPE residual on the sub-group and compare it with the overall performance. To be specific, we choose to use the data snapshots from the years 2000 (*IoM*), 2002 (*OPTN*), and 2003 (*MELD*) respectively to contrast the effect induced by data collected by different behavior policies.

Results are presented in Table 3. In addition to the epistemic uncertainty and OPE residuals of both sub-group and in population, we additionally calculate the ratio to highlight the vulnerability of the sub-group. We find the inaccuracy of OPE in this sub-group remains clearly higher than average until at least 2003, a year or so after MELD was introduced. DataCOPE successfully identifies and monitors such vulnerability with the epistemic uncertainty component, and would have been able to highlight this uncertainty to clinicians implementing the policy, allowing them to take more care in decisions for this sub-group.

**Take-Away:** *On a real-clinical dataset, we demonstrate DataCOPE accurately mirrors the behavior of the OPE residual and empowers decision-makers with deeper insights into their policy choices and evaluation. Furthermore, our examination of a particular subgroup where the OPE proves inaccurate underscores the practical value that DataCOPE can guide real-world policy deployment.*

## 5 Supplementary Results and Discussions in Appendix

In addition to the empirical verification presented in our main text, we have included several experiments and ablation studies in the appendix, along with further discussions. For the convenience of interested readers, we have summarized the key takeaways from each section of the appendix.

1. **Extended Related Work**: Appendix A further discussed related work on OPE, benchmarking algorithms, and active learning.
2. **Error Estimation for Different Policies**: Appendix B details experiments with varying noise levels in the target policy to test DataCOPE under more challenging conditions. We found that DataCOPE accurately predicts performance across OPE algorithms and stochasticity levels consistently.
3. **Data Coverage**: Appendix C examines the effect of varying data coverage, revealing that DataCOPE effectively quantifies the target policy's deviation from the behavior

policy. The epistemic uncertainty component of DataCOPE provides insight into the inherent difficulty of the OPE problem.

4. **Computational Time**: Appendix D discusses the implementation details and computational resources required for our algorithm. We emphasize that DataCOPE is lightweight, with training time typically in the order of $10^3$ seconds.

5. **Experiment Details**: In Appendix E, we further elaborated technical details of each experiment setup.

6. **Additional Experiments**: In Appendix F, we further verify the effectiveness of DataCOPE on 3 more UCI tabular datasets and the Large Language Model alignment task. Demonstrating the generality and scalability of the proposed method on real-world applications.

## 6 Conclusion

In this work, we propose DataCOPE to evaluate problems in off-policy evaluation (OPE) which is widely applicable in high-stakes real-world tasks like healthcare. DataCOPE serves as an effective proxy for the true OPE residual without access to the environment. While previous work focused on improving estimators based on offline datasets without considering the inherent difficulty introduced by the mismatch between such datasets and the target policy, we propose to re-think the problem of OPE from a data-centric perspective. The benefits of DataCOPE are demonstrated through empirical studies in both synthetic and real-world healthcare datasets.

## Acknowledgement

HS would like to acknowledge and thank the Office of Naval Research for its support through the PhD Scholarship. AJC is supported by a Microsoft Research PhD fellowship. NS is funded by the Cystic Fibrosis Trust. This work was additionally supported by the NSF (Grant number: 1722516). We would also like to thank all of the anonymous reviewers on OpenReview and our action editor Omar Rivasplata in improving the quality of this paper, alongside the many members of the van der Schaar lab, for their input, comments, and suggestions at various stages that have ultimately improved the manuscript.

## Broader Impact Statement

Our research answers the question of *when Off-Policy Evaluation (OPE) can be effective* from a data-centric perspective. The problem of OPE is generally important and has been widely applied to real-world applications like healthcare and recommendation systems. However, previous work focused on algorithm-centric development of the methods and neglected the importance of *data*. In our work, we highlight the critical role of *data* in OPE: to evaluate a certain target policy, having appropriate data that matches such a target policy evaluation is a must. Our work calls on the community of OPE research to not only focus on developing estimation algorithms but also rethink the problems from a data-centric perspective and be skeptical of using arbitrary data to evaluate arbitrary target policies.

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

## Appendix A. Extended Discussion on Related Work

### A.1 High-Level Difference: DataCOPE is a Framework for Evaluating OPE, rather than an OPE Estimator.

Although DataCOPE is situated within the general field of OPE, it differs from typical OPE papers that propose methods for solving existing OPE problems. Instead, DataCOPE functions as a type of Meta-OPE that "COPE with the Data" — it evaluates general OPE algorithms by forecasting whether the conditional expected return can be estimated accurately.

It is important to note that not all OPE problems are equal, as some can be intrinsically difficult, while others may be much easier. In our work, we focus on evaluating OPE problems (i.e., a dataset-target policy pair) rather than proposing or evaluating OPE algorithms. We present DataCOPE as a practical method and a first step toward quantifying the difficulty of OPE problems. This difficulty is an inherent quantity that relates to the mismatch between the behavior dataset and the target policy. We are motivated to introduce DataCOPE as a data-centric solution that captures the properties of the dataset and target policy.

### A.2 OPE Estimators

We provide a further discussion of the OPE literature in this section. Specifically, we connect and differentiate DataCOPE with (Tucker and Joachims, 2022; Wan et al., 2022; Jiang and Li, 2016; Thomas et al., 2015; Taufiq et al., 2022; Zhang et al., 2022; Udagawa et al., 2023).

Inverse Probability Weighting (IPW) (Precup et al., 2000; Strehl et al., 2010), a weighted sum of behavior rewards is used in estimating the value of a target policy. The Doubly Robust (DR) method (Dudík et al., 2011; Jiang and Li, 2016; Su et al., 2020a) improves the DM and IPW by leveraging the strength of both. Shrinkage techniques (Su et al., 2020a), self-normalization (Swaminathan and Joachims, 2015), and switch method (Wang et al., 2017) further address the variance issue of those estimators. The recent advance of distribution correction estimation (DICE) family (Nachum et al., 2019; Zhang et al., 2020a,b) achieve promising performance and are unified as regularized Lagrangians of the same linear program (Yang et al., 2020). Differently, in our work, we focus on the *importance of the dataset* used for those OPE estimators, which is why we emphasize the specific dependence on $\mathcal{D}$ in Equation (5).

One of the fundamental differences between our work and the literature above is that DataCOPE challenges the OPE problem itself, rather than aiming to optimize for a better OPE estimator. Our perspective is to question whether evaluating a specific target policy is reasonable, given the current dataset. This data-centric perspective is a significant contribution to the field. Instead of developing OPE algorithms without considering the data quality, we propose to evaluate whether a particular target policy can be appropriately evaluated using the available dataset. Not all target policies can be evaluated with the same level of confidence, and some may be harder to evaluate than others. DataCOPE aims to identify the easy problems and subsets from the hard ones, rather than focusing on developing OPE algorithms that perform well on some problems but not others.

To elaborate on the connections and differences between our work and the above literature in more detail:

We emphasize the importance of the data-centric perspective of OPE throughout the paper, and our contribution is not a method for data collection that treats other approaches as baselines under specific settings (budget, safety, etc.). Instead, we provide insight into the importance of a match between the dataset and the target policy. This perspective is supported by the results of (Tucker and Joachims, 2022; Wan et al., 2022), as they implicitly reveal the importance of data quality, which we focus on explicitly.

Additionally, the settings and assumptions in these works are different. Tucker and Joachims (2022) considers the problem of counterfactual predictions that is often the case in advertising. Wan et al. (2022) explores efficient data collection under safety constraints. In our work, data collection is not an option with freedom in the high-stake clinical setting. We apply DataCOPE as a demonstrative example to show how data quality affects OPE performance and use it as a proxy indicator. Our work complements this line of literature by quantifying the quality of the dataset for a given policy in terms of uncertainty.

Jiang and Li (2016) extends the doubly robust method to the sequential setting. Its counterpart in the contextual bandit setting, DR, is already used as a backbone method in verifying the universal predictive power of DataCOPE.

The work by (Thomas et al., 2015) estimates a lower bound for OPE but relies on the assumption that the behavior policy adequately covers the action space, which can not hold in high-stakes applications like clinical guidelines. In contrast, our work focuses on the problem of evaluating a target policy with an imperfect dataset, and aims to identify which policies can be accurately evaluated on which subset of the data. Thus, our approach offers a different perspective on the problem of OPE.

The works (Taufiq et al., 2022; Zhang et al., 2022) use conformal prediction techniques to improve the accuracy and reliability of OPE estimates, but they still belong to the OPE algorithm class (algorithm-centric). In contrast, DataCOPE is not an OPE algorithm but rather an evaluation method that provides a data-centric perspective on the feasibility of OPE for a given target policy and dataset. It evaluates the quality of the data and the assumptions underlying OPE, which can inform the development of OPE algorithms as well as the interpretation of their results.

Udagawa et al. (2023) examines the issue of estimator selection. In contrast, our work places greater emphasis on the notion that for certain ill-defined problems, such estimator selection may be of little help since all estimators would perform similarly poorly in comparison to when they are matched with a more suitable dataset-target policy. This highlights the fundamental difference between a data-centric approach and an algorithm-centric approach to tackling OPE problems.

### A.3 Conformal OPE and Estimator Selection

The OPE estimator selection literature primarily focuses on the algorithm-centric aspect of OPE research (Udagawa et al., 2023; Saito et al., 2021; Su et al., 2020b; Cief et al., 2024; Felicioni et al., 2024), addressing the question, "Given a specific dataset, which estimator should be used to achieve the best OPE?" In contrast, DataCOPE is designed to answer the question, "Given a target policy, is the dataset adequate for performing OPE?" Combining algorithm-centric estimator selection methods with our data-centric DataCOPE holds promise for further enhancing OPE performance.

Additionally, in certain applications, such as LLM alignment, reward modeling (i.e., the direct method in OPE terminology) is the predominant approach for model evaluation. In these cases, the application of OPE selection is restricted, whereas DataCOPE can still guide the selection of appropriate offline datasets, as demonstrated in Appendix F.2.

On the other hand, conformal OPE methods (Zhang et al., 2022; Taufiq et al., 2022) focus on incorporating uncertainty into OPE estimators, providing interval estimates. The key difference is that DataCOPE is algorithm-agnostic, allowing it to integrate seamlessly with algorithm-centric research. For example, in the context of language model evaluation, DataCOPE can first determine whether the available dataset is sufficient for reward modeling and evaluation. Subsequently, conformal OPE or other algorithms can be applied to the selected dataset.

### A.4 Benchmarking OPE Algorithms.

As with the problem of evaluating policies, actually evaluating the quality of OPE algorithms is similarly difficult. In recent years, the community has proposed various large-scale datasets for the purpose of benchmarking OPE algorithms, including Open-Bandit (Saito et al., 2020) and DOPE (Fu et al., 2021) benchmark OPE in the commercial recommendation and robotics settings. (Voloshin et al., 2019) empirically studies many current methods, and stress tests OPE algorithms in the RL setting. We note an important difference in healthcare that there are always human-based clinical guidelines, rather than machine-learning policies. Moreover, a central problem remains - given a new task for which OPE is required, there is no reliable way to estimate the quality of any value estimation - which brings us to our contribution.

### A.5 Exploration

While exploration aims to minimize long-term regret, our work is focused on minimizing the error in off-policy policy value estimation. To illustrate this, consider a clinical trial where the purpose of introducing new treatments (i.e., exploration) is to improve long-term feedback through exploration. In contrast, our work focuses on answering the question of whether, given a past or future dataset of patient information, we can minimize the estimation error of an introduced new treatment. Therefore, we believe that our work can provide valuable insights into improving the accuracy of policy value estimation and complement the exploration strategies in the literature.

### A.6 Active Learning and Active Data Collection

DataCOPE deals with a fundamentally different task from active learning (Lewis, 1995; Sebastiani and Wynn, 2000; Mussmann and Liang, 2018; Houlsby et al., 2011; Kirsch et al., 2019; Nguyen et al., 2022). The goal of active learning-based uncertainty sampling is to improve the model by selecting which samples to best label next (model-centric task). This is different from DataCOPE which is focused on an already given dataset, with the goal of characterizing the types of samples and their effect on OPE. In addition, the cost functions in acquiring new samples are always very clearly defined in active learning, yet our setting does not set explicitly trade-offs.

Our proposed metric could be extended for the purposes of data acquisition. The active learning literature has explored two main paradigms, Maximum Entropy Sampling (MES) or Uncertainty Sampling, and Bayesian Active Learning by Disagreement (BALD), which aim to query samples with the maximum predictive entropy or maximize the mutual information between the observation and the model parameters, respectively. These approaches reflect model uncertainty and are linked to our notion of epistemic uncertainty, albeit measured differently. Our work could inspire future research on how to leverage our proposed metric for more principled data acquisition in OPE settings.

Different from previous active learning works or OPE data collection methods like (Tucker and Joachims, 2022), DataCOPE is used for evaluating datasets at the same size yet with different samples to demonstrate the importance of the match between dataset and target policy in OPE problems. Those different samples are collected by different strategies: the baseline one is collected through uniform sampling, while the other one is collected according to the epistemic uncertainty.

In our experiments, we are not actively recruiting new data points that have the highest epistemic uncertainty, nor can we be able to add new features to the task to decrease aleatoric uncertainty. What we intended to emphasize in this section is that DataCOPE is a hindsight descriptive tool that compares the quality of a dataset given a certain target policy.

DataCOPE is distinct from prior active learning literature or OPE data collection methods such as (Tucker and Joachims, 2022) in that DataCOPE assesses datasets of the same size but with varying samples to emphasize the significance of matching the dataset with the target policy in OPE problems. Different sampling strategies are employed to collect these diverse samples in Section C: the baseline approach employs uniform sampling, while the other method leverages epistemic uncertainty to collect samples. The takeaway message is that: datasets are not equal in evaluating the same target policy. A lower epistemic uncertainty leads to a generally improved performance among a batch of OPE algorithms. There is no doubt that advanced data-collecting algorithms like (Tucker and Joachims, 2022) can potentially be more efficient, though might not be practical in high-stake scenarios.

To sum up the difference: in our experiments, we do not actively recruit new data points that have the highest epistemic uncertainty, nor can we add new features to the task to reduce aleatoric uncertainty. The purpose of this section is to emphasize that DataCOPE is a retrospective descriptive tool that compares the quality of a dataset given a specific target policy.

### A.7 How is Data-Centric different from Algorithm-Centric Research in OPE

In this section, we highlight the key differences between the data-centric and algorithm-centric research of OPE. We would like to start by outlining the following key differences:

1. **The Access to OPE Residual is Optional**. In DataCOPE, predicting or estimating OPE residuals is optional and not intrinsic to our methodology. It serves in the paper as an evaluation to quantify and validate the high correlation between uncertainty components and OPE residuals (i.e., the quality of dataset-target policy match). In practical applications, as demonstrated in our real-world use cases, this step is omitted.
2. **Data-Centric means Algorithm-Agnostic**. Unlike the OPE literature, which primarily focuses on identifying optimal algorithms for OPE problems, DataCOPE

adopts an algorithm-agnostic perspective. Traditional OPE studies and **estimator selection methods seek effective algorithms or combinations thereof, inherently and explicitly requiring multiple estimators.** Our approach, in contrast, emphasizes a **consistent** evaluation of the OPE problem based on the dataset and policy characteristics alone, regardless of the estimators used.

In data-centric research, such as DataCOPE, the evaluation of an OPE problem should not rely on any specific OPE estimator — such that DataCOPE only characterizes the properties of the dataset and target policy, but not any estimators. Therefore, no matter what estimators could be used, added, or removed, DataCOPE will give a consistent characterization of the OPE problem itself from a Data-Centric perspective.

Besides the difference from Saito et al. (2021) that DataCOPE does not assume specific data sources, we would like to highlight the key requirement of data-centric research is that the characterization should not rely on any specific algorithm (Udagawa et al., 2023) nor on some statistics based on a batch of algorithms (Su et al., 2020b).

We use Udagawa et al. (2023) as the representative work on estimator selection to further elaborate the conceptual differences:

Udagawa et al. (2023) does not explicitly focus on the comparison between different datasets. Importantly, as the fused estimator is still a specific algorithm, using it to estimate data quality **is not a data-centric approach**. To see this, please let us consider the fact that this predictor — if used as a data-centric characterization of the dataset and policy — its prediction will change when different estimators are used. This is, by definition, contradicting the data-centric insight.

DataCOPE steps aside from the focus on improving estimation and characterizing the dataset but uses a rigorous data-centric definition to study the quality of OPE problems. Please kindly permit us to reiterate that, this **data-centric definition does not rely on any specific algorithm** but only relies on the data and target policy, hence it is algorithm-agnostic by definition. This is different from potential heuristic approaches of using any specific estimator's performance to serve as a proxy of data quality, which are by definition algorithm-dependent and therefore not data-centric.

Looking ahead, while advancements in OPE algorithms could enhance performance (e.g., in terms of lower OPE residual in any tasks), as noted, DataCOPE aims to establish a foundational, straightforward framework for assessing the compatibility of data and policies in OPE settings. This approach does not necessitate the complex integration of future algorithmic improvements — such as using the OPE residual as a proxy of the data quality (the explanation is that, this is the best performance we can ever get). **DataCOPE provides an initial, lightweight attempt to characterize the data and target policy match in OPE problems.**

During our discussions in the revision of this manuscript, our reviewer raised a great idea of applying the recent advances presented in Cief et al. (2024); Felicioni et al. (2024) to further explore the data-centric research of OPE problems. Especially on creating synthetic data to create a quality predictor as the reviewer pointed out (Felicioni et al., 2024). Exploring data-centric OPE by combining insights from those great works provides promising future research opportunities.

## Appendix B. Using DataCOPE for Evaluating OPE Algorithms

We quantitatively demonstrate that the uncertainty components from DataCOPE can be used to estimate the OPE residuals with calibration across a range of OPE methods and therefore could be broadly used as a performance indicator when the ground truth is unknown.

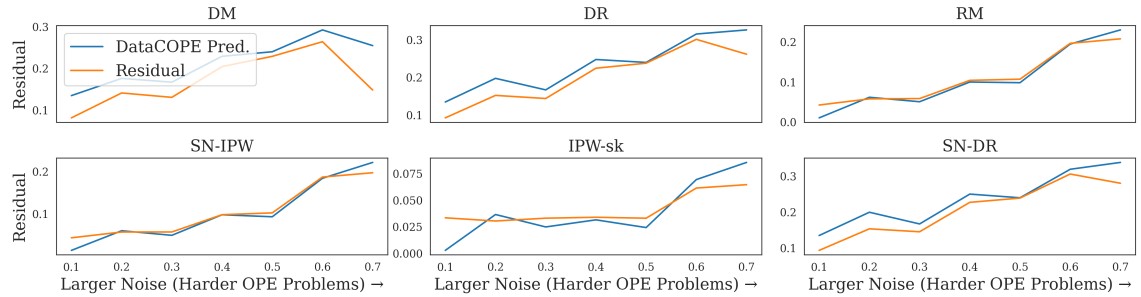

Figure 4: *DataCOPE works as an accurate proxy of the OPE residual.* DataCOPE is able to predict OPE residuals of different algorithms with calibration. Dataset: `Breast Cancer`.

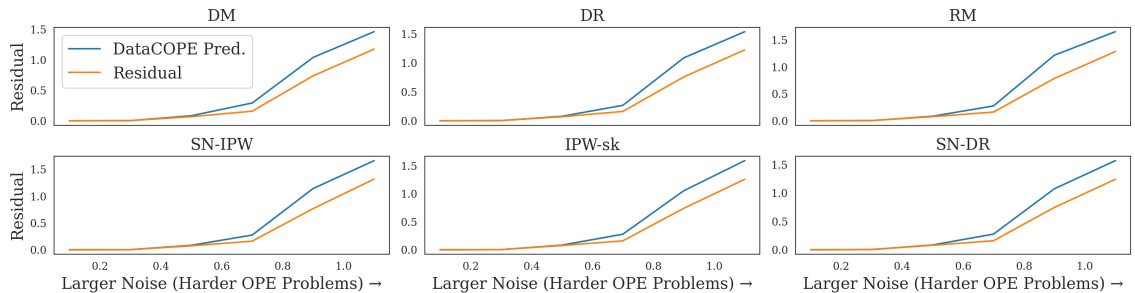

Figure 5: *DataCOPE works as an accurate proxy of the OPE residual.* DataCOPE is able to predict OPE residuals of different algorithms with calibration. Dataset: `Diabetes.`

**Experiment Settings** For each dataset, we run different OPE algorithms and are able to calculate the value estimation residual (error), comparing this to the output of DataCOPE, which has decomposed the uncertainties into aleatoric and epistemic components. The correlation between the OPE residuals and the uncertainties is then calculated and reported, noting that a strong correlation implies that the DataCOPE uncertainties have predictive power when it comes to determining if the OPE is accurate.

To test the impact of mismatched behavior and target policies, in the `Diabetes` dataset, we consider the impact of injecting different levels of noise during policy generation. With higher training noise, the policies will be forced to rely on more general features and employ simpler heuristics - in this way the relative *complexity* of the target policy compared to the behavior policy *increases* allowing the difficulty of OPE given the mismatched behavior policy and target policy to be manipulated and explored.

Specifically, we test across a range of values of complexity and add Gaussian noise with variance $[0.1, 0.3, 0.5, 0.7, 0.9, 1.1]$ during behavior policy learning. We then perform calibration to predict the OPE residuals using DataCOPE according to Equation (9): we fit model $h$ on the held-out training data, and dub such a model the calibrated OPE residual predictor, before again aiming to predict OPE residuals using our uncertainty decomposition.

**Results** Raw numerical results are reported in Table 1, which quantitatively shows the correlation between uncertainties obtained from DataCOPE and value estimation residuals of different OPE algorithms. DataCOPE in general performs well across all datasets and algorithms, with a strong correlation with the residuals.

**Take-away:** *Calibrated uncertainties decomposed by DataCOPE can be used to accurately predict the performance across a range of OPE algorithms.*

On the impact of complexity, Figure 4 and Figure 5 visualizes our experiment results, plotting both the prediction of DataCOPE, and the true OPE residual for multiple OPE methods as the complexity differential increases. DataCOPE is able to predict OPE residuals of various algorithms with high accuracy. In general, the deviations of predicted values from real residuals get higher as the complexity increases. Moreover, we empirically find DataCOPE tends to overestimate the OPE residual, lending its use as a performance lower-bound, being most suitable for high-stake application scenarios like healthcare.

**Take-away:** *As complexity differential between policies increases, the OPE performance decreases - an effect DataCOPE is able to predict and track well.*

## Appendix C. Matching Dataset and Target Policy with DataCOPE

So far we have shown that DataCOPE can tell if our OPE estimates will be any good, and for which individuals this will apply. We now move to show how the uncertainty components discovered by DataCOPE can be informative in comparing datasets for evaluating a certain target policy. We consider the scenario of the clinical setting where new policies need to be evaluated before verifying those policies in clinical trials. We show that DataCOPE can act as a performance indicator and be informative in the pursuit of efficient and accurate OPE.

**Experimental Setup** We use DataCOPE to compare two data collection strategies for OPE problems, aiming to show that the general OPE performance is aligned with epistemic uncertainty. Here, we synthetically generate a biased logged dataset by running biased and highly deterministic behavior policies $\pi_b$ for collecting those data. Specifically, we explore the *coverage* of the behavioral policy by thresholding examples by their quantile values on certain features. We remove examples with the top 60% high values at the feature named *worst concave point* in the `Breast Cancer` dataset and remove examples with top 60% high values of the feature named *log of serum triglycerides level* in the `Diabetes` dataset [1] during behavior policy generation. In the meantime, the target policy $\pi_e$ is still generated with the full dataset, and hence there exists a behavior bias on the training examples out of the threshold.

We demonstrate the effectiveness of DataCOPE by experimenting with the two most straightforward data collection strategies: uniform sampling (e.g., uniformly recruiting patients in clinical trials) and uncertainty-guided sampling — according to the principle of

---

1. these are selected as the most influential dimension for linear models to manifest the difference

*optimism in the face of uncertainty* (Kearns and Singh, 2002; Brafman and Tennenholtz, 2002) (i.e. picking those with the highest uncertainty).

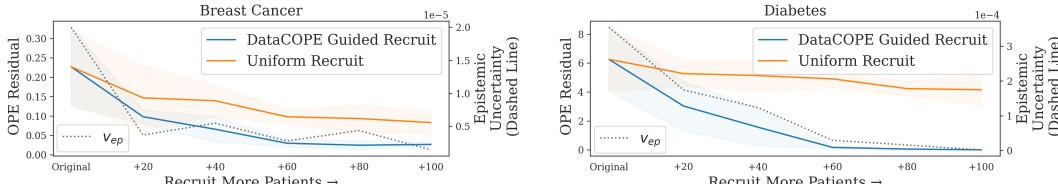

Figure 6: Increasing the number of examples according to epistemic uncertainty in the learning of $\pi_b$ clearly improves OPE performance. While uniformly sampling new examples only improves OPE a little. DataCOPE identifies which dataset or data collection strategy is better for evaluating a given target policy.

**Results** In Figure 6, we report the changes in OPE residuals as newly recruited patients are added. Therefore, the aligned x-axis indicates the size of the dataset is the same — yet their quality can be different — due to different data collection strategies and discrepancies from the behavior policy. We report the epistemic uncertainty in OPE when recruitment is guided by DataCOPE. Curves are obtained by averaging over 6 benchmark algorithms. For both datasets, we find DataCOPE is able to identify a better data collection strategy according to the epistemic uncertainty.

**Take-away:** *DataCOPE is able to identify the target policy $\pi_e$'s deviation from the behavior policy. The epistemic uncertainty component of DataCOPE informs the inherent difficulty of the OPE problem in general as well as in sub-groups. The quality of the datasets for a specific target policy evaluation task can be compared according to DataCOPE.*

## Appendix D. Implementation Details

### D.1 Code

Our code is open-sourced at `https://github.com/holarissun/Data-Centric-OPE`.

### D.2 Hardware and Training Time

We experiment on a machine with 2 TITAN X GPUs and 32 Intel(R) E5-2640 CPUs. In general, the computational expense of DataCOPE is cheap. With our PyTorch-based implementation, decomposing the uncertainty components with DataCOPE using 100 neural network ensembles takes approximately 10 minutes to run. OPE algorithms take 2 to 10 minutes to run depending on their complexity and can be accelerated by using GPUs.

### D.3 OPE Algorithms

Our implementation of OPE algorithms is based on the implementation of (Saito et al., 2020), open-sourced at `https://github.com/st-tech/zr-obp/tree/master/obp/ope`.

Different from (Saito et al., 2020), we do not assume the access of the behavior policy $\pi_b$, hence we build estimators of $\pi_b$ based on the dataset by training classification or regression models with neural networks if a parameterized behavior policy is needed. e.g., in the case of IPW.

## D.4 Hyper-Parameters for Neural Approximators

The hyper-parameters for neural-network-based approximators we used for MDN are reported in Table 4.

Table 4: Hyper-parameters of neural network approximators

| Hyper-param | Choice |
|---|---|
| Network Layer | 3 |
| Activation Function | ReLU |
| Hidden Units | 64 |
| Training Epoch | 100 |
| Optimizer | Adam |
| Learning Rate | 1e-3 |

## D.5 Hyper-Parameters for DataCOPE

In DataCOPE, there are two hyper-parameters to be specified: the number of ensemble models used and the number of Gaussians in the MDN model in regression tasks. DataCOPE is robust to both hyper-parameters in our empirical study.

In our experiments, we find using 100 ensemble models is computationally affordable on tabular data. As it takes less than 10 minutes to run. In the meantime, we find using 10 ensemble models can provide decent performance.

The main objective of introducing MDN is to be able to quantitatively decompose the uncertainty in regression settings. Therefore, while the number of Gaussian mixtures used should depend on the inherent structure of data distribution, whether or not capturing the exact distribution is of less importance than being able to capture the uncertainty components. The number of Gaussians we use in our experiment is set to be 3 for all datasets. We empirically find using $3, 5, 10$ mixture of Gaussians does not clearly affect the performance.

## Appendix E. Experiment Details

### E.1 Calibration

In practice the OPE residual $\bar{\xi}$ is always infeasible, such a difficulty leads us to introduce a practical substitute that estimates $\bar{\xi}$ without relying on that infeasible ground-truth policy value. We elaborate on the calibration step in this section.

We leveraged a cross-validation type method in the calibration step. To conform with typical procedures in supervised learning, we hold out a portion of the training data, where we have knowledge of their accurate instance-wise value (i.e., the return of the context-action pairs). By doing this, we are able to break down the uncertainties of the instance-wise value prediction. Subsequently, we use the reserved ground-truth value to calculate the OPE residuals of each OPE algorithm as the objective, and uncertainties as the input of a regression model. Our implementation employs the bootstrap sampling method.

It is also worth mentioning that such a calibration step is optional, and DataCOPE can evaluate OPE problems without calibration:

In fact, our contribution and novelty of DataCOPE do not rely on such a calibration step. This is demonstrated in our experiments: The difficulty of OPE problems is correlated with our decomposed uncertainties. Such a decomposition, without calibration, is useful in that (1) it permits a comparison among datasets to answer the question of "which dataset is most appropriate in evaluating a certain given policy" (Section C) (2) it permits a hindsight interpretation of clinical guideline deployment and vulnerable group identification. (Section 4.2)

## E.2 Synthetic Dataset

**Policy Generation**    Following standard procedures in the OPE literature (Chu et al., 2011; Li et al., 2012; Agrawal and Goyal, 2013), we adapt supervised learning datasets into example logged bandit datasets where we know the true underlying generative process.

We use linear models for the behavior policy $\pi_b$, which is trained on the dataset examples $\{(x_i, y_i)\}_{i=1}^N$. Given context $x_i$ with corresponding labels $y_i$ we learn a policy to generate actions by minimizing the expected negative log-likelihood

$$\mathbb{E}_{x,y}\big[\text{NLL}(\pi_b(x_i), y_i)\big], \tag{10}$$

under a predictive Gaussian/Bernoulli distribution for regression/classification respectively.

We then evaluate $(x_i, a_i)$ with the help of ground-truth labels $y_i$. For classification tasks, the reward of action $a$ is given as $r_i^a = \mathbf{1}(a_i = y_i)$, where $\mathbf{1}$ is the indicator function; while for regression tasks, the reward of action $a_i$ is given by the coefficient of determination of the prediction. i.e., $r_i^a = 1 - \frac{u}{v}$, where $u$ is the residual sum of squares $u = ||y - a||_2^2$ and $v$ is the total sum of squares $v = ||y - \bar{y}||_2^2$, $\bar{y}$ denotes the mean of $y$. In this way, we can generate $\{(x_i, a_i, r_i^a)\}_{i=1}^N$ tuples as our dataset containing $N$ examples.

**Target Policy**    We also generate target policies using neural network models trained on the *same* training data $\{(x_i, y_i)\}_{i=1}^N$, but with injected noise (that will depend on the particular experiment, as $\pi_e$. The ground-truth performance of policy $\pi_e$ is then given by $r_j^b = \mathbf{1}(b_j = y_j)$, where $b_j = \pi_e(x_j)$ - importantly this is available to us and thus allows for the evaluation of the quality of the OPE.

In particular, the off-policy evaluation problem is by definition estimating $\mathbb{E}[r_j^b]$, for $j = 1, ..., N$. As we have access to $y_j, j = 1, ..., N$, we can quantitatively evaluate our method and compare results with the ground-true policy values.

## E.3 Organ Transplant

**Data Description and Logged Contextual Bandit Formalism.**    We examine data from the Organ Procurement & Transplant Network (OPTN) (Leppke et al., 2013), which includes information on patients registered for liver transplants from 1995 to 2020. Our focus is on the policy of matching organs that become available to patients who are waiting for a transplant.

For each decision, the set of potential patients (action space) includes those on the waitlist at the time the organ becomes available, and the information considered for each patient (context) includes both the organ and the patient's characteristics.

**Data Preparation.** The OPTN dataset includes 308,912 patients who were either waiting for or had received a liver transplant. We eliminated patients who hadn't received a transplant, were under 18 or had a donor under 18, or had missing data for certain variables, leaving us with 31,045 patients. Consequently, we consider 8 features in total: *ABO Mismatch, Age, Creatinine, Dialysis, INR, Life Support, Bilirubin,* and *Weight Difference.*

The focus of our experiments is on patients who received organs prior to 2005, allowing us to divide the data into distinct phases based on evolving allocation guidelines. Specifically, 512 patients were allocated organs under the Institute of Medicine recommendations before 2000, 1969 patients under the OPTN Final Rule between 2000 and 2002, and 2515 patients after the implementation of MELD between February 26th, 2002 and May 1st, 2003. Those data serve as the training set, with additional data containing 3914 patients collected between May 1st, 2003 and January 1st, 2005 used to evaluate OPE for MELD and DataCOPE's estimation of MELD.

We streamline the data following (Hüyük et al., 2022) and select patients out of their contemporary patients from the transplant waitlist for an organ to be allocated to. Hence, the dataset will be formalized as a logged contextual bandit task. For every coming organ, the allocation policy selects one from the waiting patients to allocate the organ and receives the patients' survival time as the reward.

## Appendix F. Additional Experiments

### F.1 Additional Results on the UCI Datasets

In order to validate that DataCOPE is effective and powerful, we further validate DataCOPE on other UCI datasets beyond the `Diabetes` (1-dim integer regression ranging from value $[25, 346]$) and `Breast Cancer`(2-class classification). We experiment with synthetic contextual bandit dataset generated from both classification tasks `Digits, Wine` (10-class, 3-class classification) and regression task `Boston` (1-dim regression ranging from value $[5, 50]$). Results are presented in Table 5. On all datasets, we find DataCOPE is able to act as a well-performing proxy of the OPE residual.

### F.2 Additional Results on Large Language Model Alignment (Reward Modeling)

To further verify the scalability of DataCOPE in large-scale experiments, we further verify the effectiveness of DataCOPE in evaluating preference datasets in the reward modeling step of Large Language Model alignment.

**Background: Reinforcement Learning from Human Feedback with Offline Annotations** The alignment of LLMs is essential for their safe and effective deployment in real-world applications. The technique of reinforcement learning from human feedback (RLHF) (Christiano et al., 2017; Ouyang et al., 2022) plays a crucial role in alignment. In RLHF, offline preference annotations are used to generate a reward model Rafailov et al. (2024); Zhao et al. (2023); Yang et al. (2024b); Dong et al. (2023); Sun et al. (2023); Azar et al. (2024); Sun (2023); Munos et al. (2024); Chan et al. (2024), which can perform off-policy evaluation and can be used in the later stage of policy improvement (i.e., LLM fine-tuning).

Table 5: DataCOPE is able to predict the difficulty of various OPE algorithms under various settings. The correlation between DataCOPE's two uncertainties and OPE residuals under different settings is high. The ablation studies show that uncertainty decomposition is important in predicting OPE performance. (Reported numbers: correlation for DataCOPE rows, and for the ablation studies we report the performance difference compared with DataCOPE, **higher is better**)

| Dataset | Ablations | DM | DR | RM | SNIPW | IPWsk | SNDR |
|---|---|---|---|---|---|---|---|
| Wine | DataCOPE | 0.801 | 0.873 | 0.915 | 0.918 | 0.829 | 0.895 |
| | w/o $v_{al}$ | 0.797 ($\downarrow$ 0.004) | 0.855 ($\downarrow$ 0.018) | 0.677 ($\downarrow$ 0.238) | 0.725 ($\downarrow$ 0.193) | 0.653 ($\downarrow$ 0.176) | 0.850 ($\downarrow$ 0.045) |
| | w/o $v_{ep}$ | 0.752 ($\downarrow$ 0.049) | 0.846 ($\downarrow$ 0.027) | 0.878 ($\downarrow$ 0.037) | 0.898 ($\downarrow$ 0.02) | 0.810 ($\downarrow$ 0.019) | 0.887 ($\downarrow$ 0.008) |
| | w/o Decomposition | 0.786 ($\downarrow$ 0.015) | 0.870 ($\downarrow$ 0.003) | 0.825 ($\downarrow$ 0.09) | 0.856 ($\downarrow$ 0.062) | 0.772 ($\downarrow$ 0.057) | 0.895 ($\downarrow$ 0.0) |
| Digits | DataCOPE | 0.912 | 0.907 | 0.963 | 0.996 | 0.899 | 0.957 |
| | w/o $v_{al}$ | 0.688 ($\downarrow$ 0.224) | 0.687 ($\downarrow$ 0.22) | 0.959 ($\downarrow$ 0.004) | 0.968 ($\downarrow$ 0.028) | 0.800 ($\downarrow$ 0.099) | 0.831 ($\downarrow$ 0.126) |
| | w/o $v_{ep}$ | 0.612 ($\downarrow$ 0.3) | 0.612 ($\downarrow$ 0.295) | 0.963 ($\downarrow$ 0.0) | 0.933 ($\downarrow$ 0.063) | 0.745 ($\downarrow$ 0.154) | 0.769 ($\downarrow$ 0.188) |
| | w/o Decomposition | 0.671 ($\downarrow$ 0.241) | 0.669 ($\downarrow$ 0.238) | 0.961 ($\downarrow$ 0.002) | 0.961 ($\downarrow$ 0.035) | 0.788 ($\downarrow$ 0.111) | 0.817 ($\downarrow$ 0.14) |
| Boston | DataCOPE | 0.805 | 0.787 | 0.786 | 0.779 | 0.776 | 0.787 |
| | w/o $v_{al}$ | 0.781 ($\downarrow$ 0.024) | 0.725 ($\downarrow$ 0.062) | 0.726 ($\downarrow$ 0.06) | 0.722 ($\downarrow$ 0.057) | 0.731 ($\downarrow$ 0.045) | 0.719 ($\downarrow$ 0.068) |
| | w/o $v_{ep}$ | 0.734 ($\downarrow$ 0.071) | 0.682 ($\downarrow$ 0.105) | 0.669 ($\downarrow$ 0.117) | 0.672 ($\downarrow$ 0.107) | 0.682 ($\downarrow$ 0.094) | 0.680 ($\downarrow$ 0.107) |
| | w/o Decomposition | 0.735 ($\downarrow$ 0.07) | 0.683 ($\downarrow$ 0.104) | 0.669 ($\downarrow$ 0.117) | 0.673 ($\downarrow$ 0.106) | 0.682 ($\downarrow$ 0.094) | 0.681 ($\downarrow$ 0.106) |

The reward modeling step is a special type of the Direct Method in the off-policy evaluation literature — building on top of the Bradley-Terry model Bradley and Terry (1952).

In practice, collecting the offline preference annotation is costly Xiong et al. (2023); Guo et al. (2024); Tang et al. (2024), therefore, how to maximally utilize existing offline (off-policy) annotation datasets is an important question. Formally, when trying to optimize the performance of an LLM $\ell$ in a task $\mathcal{T}$ (e.g., helpful chatbot), there are multiple open-sourced preference annotations released by different research teams. Those datasets, while sharing the same prompt set (i.e., queries) $\{x_i\}_{i \in [N]}$, have different preferred and dispreferred responses as they are generated by different LLMs. For instance, different research teams may start with different foundation models like LLaMA3 Touvron et al. (2023) or Gemma Team et al. (2024); use different model sizes like 2b, 7b, or 8b; and fine-tune those foundation models on different datasets before conducting preference annotations Saeidi et al. (2024). Those different choices will lead to different offline preference dataset $\mathcal{D}_k = \{x_i, y_{i,k}^+, y_{i,k}^-\}_{i \in [N]}$.

In this experiment, we use DataCOPE to answer the following question: given an LLM $\ell$, how to decide which dataset (and hence their corresponding reward models) to use to optimize its generation?

**Motivating Example: Less could be More in Reward Modeling** In this section, we empirically demonstrate the usage of all offline datasets may not be an ideal approach to off-policy evaluation in large-scale real-world applications such as LLM alignment.

In LLM alignment, the user feedback is always provided in the format of preference annotations on some off-policy generations. For example, the Anthropic-HH dataset consists of the annotated responses generated by their 52B model (Bai et al., 2022a). In practice, when we want to align smaller open-sourced models, it will suffer from the off-policy annotation problem. When accessing different annotation sources, using all of those annotations to build

a reward model may not be the optimal choice. This is aligned with recent advances in the RLHF research (Zhou et al., 2023; Yang et al., 2024a).

To empirically verify less can be more in reward modeling, we experiment with the Anthropic-Helpful dataset, and show the results in Figure 7: while under low data regime, combining the existing offline annotation and online annotation can achieve better performance than using the pure online data, when the number of online annotation increases, it outperforms the naive combination of all datasets. Such a conclusion holds on different LLMs and different Best-of-N numbers. Next, we will apply DataCOPE to demonstrate how to use uncertainty to predict whether reward models built with offline datasets can give accurate reward predictions.

**Experiment Setups** To simulate consistent annotation at a low cost, we use the open-sourced Mistral7b-RM Dong et al. (2023) that ranks top on the RewardBench leader board Lambert et al. (2024) as the golden reward model to generate annotations. **We note that in practice, a perfect golden reward model does not always exist for general tasks — human annotators always serve as a proxy for this ideal model, providing their preferences over different responses.** We use the Anthropic Helpful Bai et al. (2022b) dataset as the prompt and consider the following 9 different preference datasets and reward models: the usage of 3 base models (Gemma2b, Gemma7b, Llama3-8b), and 3 fine-tuning setups (no fine-tuning, fine-tuned on the positive sample, fine-tuned on expert-generation).

Now to optimize the generation of a specific language model $\ell$ at test time, we aim at using DataCOPE to identify which offline dataset out of the 9 datasets above is the most suitable for conducting off-policy evaluation on $\ell$, i.e., evaluating the quality of the $\ell$-generated responses. To quantify the OPE ability, we generate 2 responses $a_1, a_2 \sim \ell(x_i)$ for each prompt $x_i$. The 9 reward models can give different value estimations of those responses $a_1, a_2$, and we use the golden reward model to evaluate their success rate in making a correct prediction of the preferred response. On the other hand, DataCOPE can provide uncertainty information when those reward models make value predictions. Given the fact each pairwise comparison is only made once, and the golden reward model's annotations are deterministic, we only consider the epistemic uncertainty component in this experiment.

To be able to perform efficient ensemble in uncertainty estimation in DataCOPE, we use the Gemma2b model to generate embeddings of different responses, and then use the gradient-boosting tree method Chen and Guestrin (2016) to build multiple reward models heads for uncertainty estimation. Since the computational bottleneck is on LLM embedding generation, which only needs to be performed once, DataCOPE is still a lightweight method in LLM alignment applications.

**Results** Table 6 shows the results. We observe that the uncertainties in making predictions are highly correlated with the reward modeling error rate, hence we can identify the most suitable offline dataset and reward models for different LLMs with DataCOPE. Further, we experiment with a larger dataset size and find that increasing the number of training samples does not always improve the reward modeling performance, but it consistently reduces the uncertainty in making predictions and improves the correlation between the uncertainty prediction and true errors.

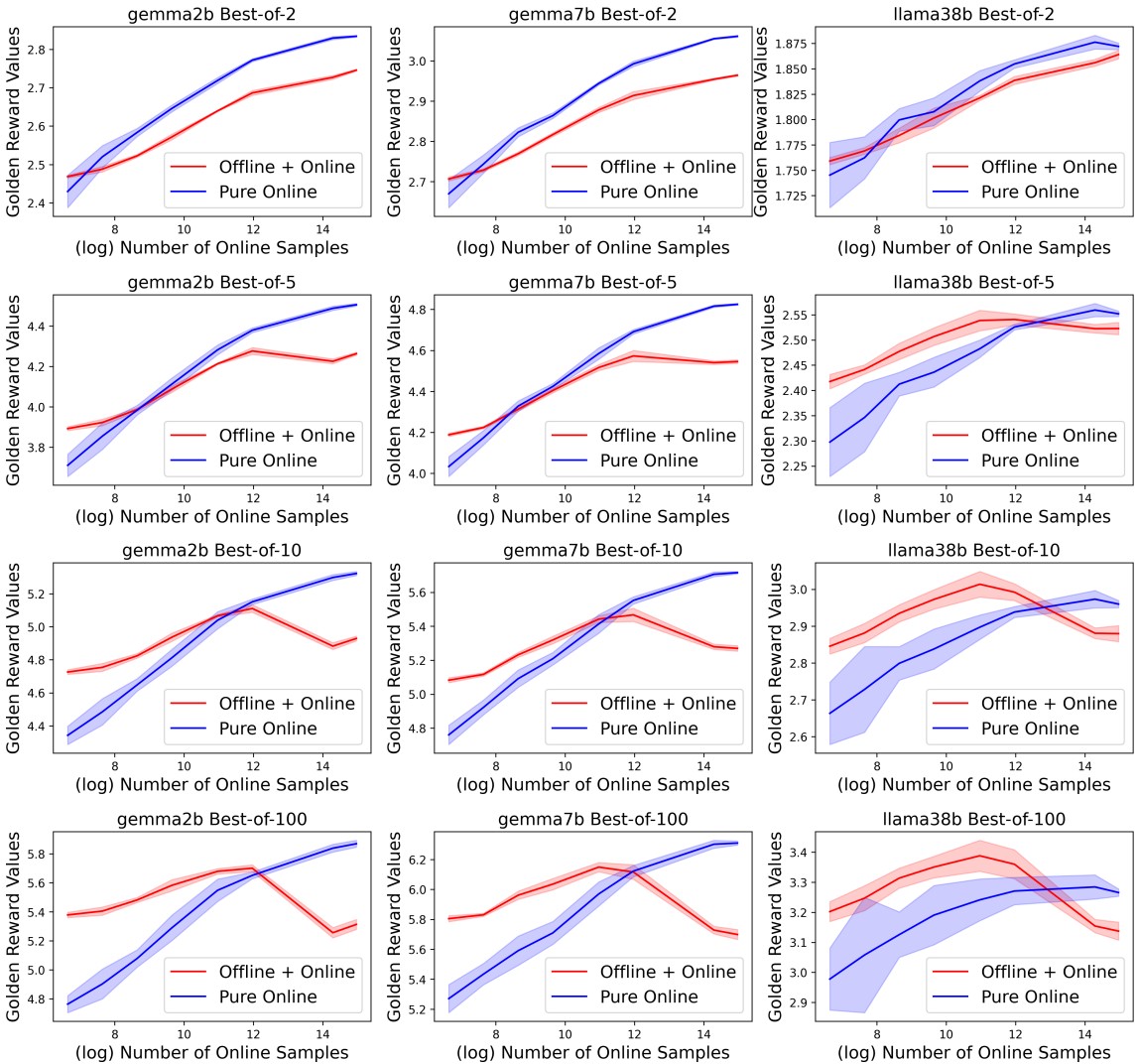

Figure 7: *Less can be more in reward modeling.* Under a low data regime, combining the existing offline annotation and online annotation can achieve better performance than using the pure online data, when the number of online annotations increases, it outperforms the naive combination of all datasets. Such a conclusion holds on different LLMs and different Best-of-N numbers. The results reported are from 5 runs.

## Appendix G. Discussion on Limitations and Future Work

In our work, we introduce DataCOPE as a Data-Centric solution for evaluating OPE algorithms. However, it is important to note that while DataCOPE serves as an OPE performance proxy, it does not directly improve the performance of OPE algorithms. It addresses the question of the difficulty of OPE problems but does not provide solutions for solving these OPE problems more accurately (without obtaining more in-distribution data). Moreover, our exploration of dataset-policy matching is conducted in a post-hoc manner

Table 6: We experiment with 3 different LLMs as the target policies and use DataCOPE to predict whether reward models built with offline datasets can give accurate reward predictions. We observe that the uncertainties in making predictions are highly correlated with the reward modeling error rate, hence we can identify the most suitable offline dataset and reward models for different LLMs with DataCOPE. Further, we find that increasing the number of training samples (dataset size) does not always improve the reward modeling performance, but it consistently reduces the uncertainty in making predictions and improves the correlation between the uncertainty prediction and true errors.

| Model | | $D_1$ | $D_2$ | $D_3$ | $D_4$ | $D_5$ | $D_6$ | $D_7$ | $D_8$ | $D_9$ | Corr. |
|---|---|---|---|---|---|---|---|---|---|---|---|
| **1x Preference Data** | | | | | | | | | | | |
| Gemma2b | Uncertainty | 0.0817 | 0.0764 | 0.0821 | 0.0728 | 0.0751 | 0.0773 | 0.0663 | 0.0649 | 0.0664 | 0.8941 |
| | Error % | 0.0881 | 0.0925 | 0.0986 | 0.0724 | 0.0711 | 0.0764 | 0.0593 | 0.0615 | 0.0659 | |
| Gemma7b | Uncertainty | 0.0827 | 0.0775 | 0.0828 | 0.0754 | 0.0765 | 0.0786 | 0.0695 | 0.0640 | 0.0691 | 0.8957 |
| | Error % | 0.1017 | 0.1095 | 0.1147 | 0.0820 | 0.0855 | 0.0912 | 0.0812 | 0.0606 | 0.0720 | |
| Llama3-8b | Uncertainty | 0.0868 | 0.0838 | 0.0871 | 0.0718 | 0.0747 | 0.0759 | 0.0718 | 0.0675 | 0.0684 | 0.9023 |
| | Error % | 0.1018 | 0.1005 | 0.1092 | 0.0870 | 0.0892 | 0.0944 | 0.0709 | 0.0692 | 0.0579 | |
| **3x Preference Data** | | | | | | | | | | | |
| Gemma2b | Uncertainty | 0.0658 | 0.0639 | 0.0658 | 0.0581 | 0.0595 | 0.0603 | 0.0528 | 0.0518 | 0.0536 | 0.9603 |
| | Error % | 0.0864 | 0.0886 | 0.0908 | 0.0702 | 0.0798 | 0.0794 | 0.0589 | 0.0654 | 0.0615 | |
| Gemma7b | Uncertainty | 0.0662 | 0.0641 | 0.0663 | 0.0590 | 0.0598 | 0.0614 | 0.0546 | 0.0512 | 0.0542 | 0.9875 |
| | Error % | 0.1082 | 0.1095 | 0.1121 | 0.0842 | 0.0894 | 0.0960 | 0.0742 | 0.0659 | 0.0729 | |
| Llama3-8b | Uncertainty | 0.0681 | 0.0667 | 0.0684 | 0.0591 | 0.0598 | 0.0612 | 0.0559 | 0.0538 | 0.0550 | 0.9267 |
| | Error % | 0.1023 | 0.1014 | 0.1075 | 0.0766 | 0.0896 | 0.1001 | 0.0731 | 0.0666 | 0.0596 | |

(i.e., passive). While it highlights the differences and importance of dataset-target policy alignment, it does not directly enable active data collection (i.e., active). This limitation is inherent to OPE problems themselves, such as in healthcare applications where collecting data on specific patient types is not feasible. However, to evaluate a certain target policy, comparison between available datasets is feasible and important.

In future work, several potential directions are promising: the most straightforward extension is to combine DataCOPE with off-policy improvement, however, this requires a trade-off between policy evaluation confidence and potential policy evaluation performance; similarly, DataCOPE can be combined with uncertainty-based exploration algorithms and be extended to the online setting, where data-centric policy learning can be studied; finally, DataCOPE can be linked with offline RL, where multiple-step decision-making policies need to be evaluated and improved. We believe that all of these directions are essential and warrant further investigation, though they are beyond the scope of this paper.

## Appendix H. Supplemental Results

In Figure 8, we present the unaveraged version of the data from Figure 2. The results indicate a strong correlation between the uncertainty components estimated by DataCOPE and the residuals from various OPE (Off-Policy Evaluation) methods.

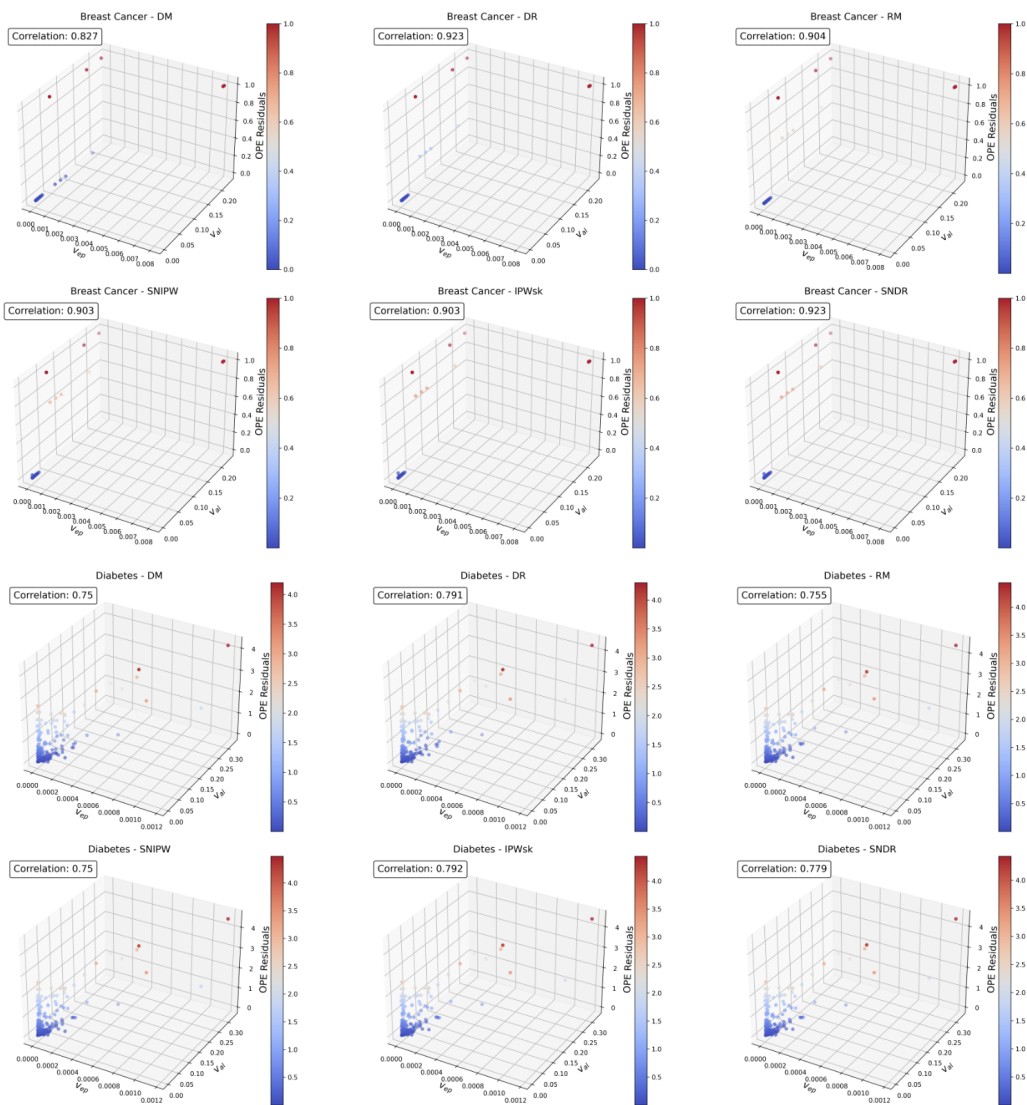

Figure 8: *DataCOPE provides accurate proxies of the OPE residual, regardless of what OPE method is used.* As supplemental results for Section 4.1, we provide the non-averaged version of Figure 2. We can conclude from the results that the uncertainty components provided by DataCOPE is highly correlated to the OPE residuals for all different OPE methods.

