# OpenReview forum: "When is Off-Policy Evaluation (Reward Modeling) Useful in Contextual Bandits? A Data-Centric Perspective"
_DMLR — Accepted by DMLR_

### Review · Reviewer_BmKU · 2024-08-09

**Recommendation:** 3
**Confidence:** 2

**Summary Of Contributions:**

The paper introduces DataCOPE, a novel framework that evaluates Off-Policy Evaluation (OPE) tasks from a data-centric perspective, focusing on predicting instance-wise evaluation errors. Unlike traditional OPE methods that focus on estimator improvements, DataCOPE assesses whether OPE problems are well-defined by decomposing uncertainties into aleatoric and epistemic components. Through empirical studies on Breast Cancer and Diabetes datasets, as well as a real-world case study on organ transplant policies, the authors demonstrate DataCOPE's effectiveness in predicting OPE residuals and identifying vulnerable sub-groups with high uncertainty. The paper also includes ablation studies to validate the importance of each component within DataCOPE, advocating for a shift in OPE research towards data-centric challenges.

**Strengths:**

Significance of the Contribution: The introduction of DataCOPE offers a novel, data-centric approach to Off-Policy Evaluation (OPE), shifting the focus from improving estimators to addressing inherent data challenges. This new perspective makes a significant contribution to the field, particularly in high-stakes applications like healthcare.

Relation to Prior Work: The paper builds effectively on existing OPE research by introducing the decomposition of uncertainties into aleatoric and epistemic components, offering a deeper understanding of OPE challenges.

Relevance to the Broader Research Community: The relevance of this work is highlighted by its application to real-world datasets, with DataCOPE's potential applicability across various domains making it valuable to a range of researchers and practitioners.

Quality of the Research: The empirical validation on both synthetic and real-world datasets demonstrates that DataCOPE is an effective evaluation approach. The use of ablation studies and correlation analyses further supports the validity of its findings.

Clarity of Paper: The paper is generally well-organized and presents its ideas clearly, making it accessible to readers. The structure, moving from methodology and simulations to practical applications, helps in understanding the significance of the contributions.

Ethical and Social Implications: DataCOPE's ability to identify vulnerable sub-groups and improve policy evaluations holds potential for positive social impact.

**Audience:**

Yes

**Broader Impact Concerns:**

The method proposed in the paper does not have ethical issues.

**Claims And Evidence:**

Yes.

**Datasets And Benchmarks:**

There is sufficient detail on data collection and organization.

**Extended Submissions:**

Not available.

**Limitations:**

see **Requested Changes**

**Requested Changes:**

1. Clarification of Method Abbreviations: The method abbreviations in the tables are not clearly defined in the main text. These should be introduced and explained the first time they appear.
2. Figure 2 Misrepresentation: The description in Figure 2 mentions color intensity changes, but the actual figure shows a color shift from blue to red. The authors should rephrase this for clarity.
3. Scale Differences in Figure 2: The significant difference in scales between the x and y axes in Figure 2 might affect the interpretation of the results. The authors should discuss whether this could impact the estimation algorithm’s performance.
4. Impact of Data Scale: While the paper claims that the method is data-centric and not influenced by the algorithm, it would be beneficial to simulate and discuss the impact of varying data scales to provide a deeper analysis.
5. Minimal Differences Between Decomposition and $v_{ep}$: In several tables, the differences between the Decomposition and v_ep methods are minimal, with identical standard deviations in the appendix. The authors should investigate and explain this occurrence.
6. Inconsistencies in Appendix Table 5: The RM and SNIPW algorithms for the Digits dataset show noticeable discrepancies compared to other methods. The authors should address when and why the proposed method might fail, and why some algorithms manage to overcome data-related issues.
7. Comparison with Other Evaluation Methods: The paper would benefit from a comparison of the proposed method with other evaluation methods to better position its contributions.

**Strengths And Weaknesses:**

### Strengths:
1. The paper introduces a new perspective in OPE research by focusing on data-centric challenges rather than solely on estimator improvements, bringing a fresh dimension to the field.
2. The breakdown of uncertainties into aleatoric and epistemic components is a contribution, offering insights into OPE challenges and guiding data collection efforts.
3. An effective evaluation proxy for OPE and performance prediction on instance-wise value estimation, with promising potential for broader applications.

### Weaknesses:
1. Some descriptions in the paper are not rigorous enough, and certain parts need clearer explanations.
2. The ablation experiments lack results from additional dimensions that could provide a more comprehensive analysis.
3. The proposed method may fail on certain datasets, but there is a lack of deeper exploration into these potential limitations.
4. The paper falls short in considering the applicability of the method to larger-scale data and more complex scenarios.

---

### Review · Reviewer_8EK5 · 2024-08-14

**Recommendation:** 3
**Confidence:** 2

**Summary Of Contributions:**

1. The authors introduce DataCOPE, an innovative data-centric framework for evaluating OPE problems. Unlike traditional OPE approaches that focus on improving algorithms for value estimation, DataCOPE evaluates the inherent difficulty of OPE problems by considering the quality and characteristics of the dataset.
2. DataCOPE can serve as a proxy for the true OPE residual without needing access to the real environment or the true value of the target policy.
3. The framework allows for the identification of sub-groups within the dataset where OPE might be inaccurate, helping to pinpoint areas where data collection or model improvement efforts should be focused.

**Strengths:**

see **Strengths And Weaknesses**

**Audience:**

Yes

**Broader Impact Concerns:**

No potential negative social impact.

**Claims And Evidence:**

Yes.

**Datasets And Benchmarks:**

Yes.

**Extended Submissions:**

Yes.

**Requested Changes:**

1. In the caption of the Figure 2, the term "Shallow color" in the second line seems to be a typo. The color bar suggests that red indicates higher residuals.

**Strengths And Weaknesses:**

### Strengths
1. The introduction of DataCOPE is a novel and significant contribution to the field of OPE. The focus on data-centric evaluation rather than solely algorithm-centric approaches is a refreshing and impactful shift.
2. The decomposition of uncertainty into aleatoric and epistemic components allows for a nuanced understanding of the sources of prediction error. This separation provides valuable insights for improving model accuracy and data collection strategies.
3. The authors provide extensive empirical validation of DataCOPE across multiple datasets, including synthetic and real-world healthcare data. The high correlation between the uncertainties and OPE residuals demonstrates the practical utility and accuracy of DataCOPE.
4. DataCOPE is able to identify vulnerable sub-groups that are more likely to suffer from inaccurate OPE estimation.
5. The paper includes a thorough analysis of the impact of different uncertainty components and data collection strategies.

### Weaknesses
1. The optional nature of the calibration step in the DataCOPE algorithm may lead to confusion about its necessity. Clarifying when and why calibration should be used could improve the practical usability of the framework.
2. The scalability and efficiency of DataCOPE when applied to very large datasets or real-time applications are not extensively discussed. Addressing these concerns and suggesting potential optimizations could improve the framework's applicability in practice.

---

### Review · Reviewer_RDoP · 2024-08-26

**Recommendation:** 3
**Confidence:** 1

**Summary Of Contributions:**

The paper considers an important challenge of understanding how reliable is the data towards off-policy evaluation in contextual bandit settings. To do so, the proposed method first estimates the aleatoric (using mixture density network) and epistemic uncertainty (using ensembles for the reward model. Then the method learns a linear predictor that uses the estimted epistemic and aleatoric uncertainty to predict the mean squared error for off-policy evaluation. The idea being that once such a predictor is available, it can be used to determine whether a dataset can be used to reliably do OPE for a desired target policy or not.

--
Post rebuttal update: updated score.

**Strengths:**

see above

**Audience:**

Yes

**Claims And Evidence:**

Claims:
Method agnostic
individualized evaluation
evaluate data collection strategy

**Datasets And Benchmarks:**

-

**Extended Submissions:**

-

**Limitations:**

see above

**Requested Changes:**

Questions:

--------------
Method:

1. The method is neither empirically demonstrated nor it looks like as if it will be able to address horizon>1. As such, it might be important to make it clear in the title/abstract that the work is focused on contextual bandits.

2. Eq 6: it is unclear what the expectation is over? Is it over a random initialization over $\theta$? Or for $\theta$ sampled from some posterior $P(\theta|D)$?

3. Eq 6: I get the spirit of the equation, but I am not sure if this is technically precise. It seems to be a proportionality by definition; how do we know that the error will have the same proportionality scaling constant? (Further, should it be $\forall \hat V_A$ or $\forall A$?)

4. How does Eq 9 use Eq 3? We do not usually have access to V(\pi) or monte-carol estimates of it to estimate MSE.
Maybe I dont understand what this sentence means "fit it with a group of held-out training data from D where true residual can be obtained without the environment:" How do you get true residual?

5. Eq 9: $\bar \xi$ is with respect to which estimator $\hat V$? Does this use a single estimator or. multiple estimators?

6. The proposed method completely ignores the bias aspect for both the OPE algorithms, and the reward model being built to assess the difficulty of the problem. On the algorithm side, Eq 3 becomes weird when some estimators are consistent, and some are not (which is often the case in practice). On the reward model side, aleatoric and epistemic uncertainty would be more useful when the reward model is itself in some sense accurate/consistent (for e.g., a reward model that outputs 0 always, uncertainty measures around it would be useless for any downstream tasks). Discussing these things carefully can make the draft much better.

7. In Eq 6, h was explicitly a policy and dataset dependent function. In Eq 9, it is no longer explicitly dependent on a policy related statistic. It only depends on the statistics of the dataset. How can h be used to evaluate the utility of the dataset towards OPE for different target policies?

----------------
Experiments:

8. What are the dim of the target/action for the regression domains, and the number of classes for the classification domains?

9. IIUC, Reward function for experiments are deterministic, thereby completely removing the aspect of aleatoric uncertainty?

10. I am not sure if I understood Table 1 properly. What does each column represent? Which OPE algorithms were used during training (Eq 3 and Eq 9) and which were used during evaluation for each of the columns? I think, ideally, there should be a single $h$ model, trained without the knowledge of the estimator being used to validate the utility of $h$.

11. Figure 2 couldnt have been more confusing :(
﻿
﻿Is it possible to replot these using 4 figures. 2 for each domains, ep and al independently, where the y-axis is the OPE residual. It is not clear how to evaluate the correlations using colors, when both of them are plotted simultaneously.

12. Figure 2: Which OPE method is being used to get the residual? Does the result (at instance level) stay consistent across different OPEs

13. I am quite confused with the experimental setting for the real-world case study (disclaimer: I have no prior idea about this domain)
- What does it mean to add 9 other contemporary patients?
- what is the dimensions of the state and action sets?﻿
- How deterministic is the evaluation policy?

14. All the following ablations are critical to justify the utility of the proposed method: Different OPE algorithms (training time + evaluation time), error estimation for different policies (deterministic, stochastic, optimal, sub-optimal), different kinds of data coverage (with support overlap, without support overlap). I understand that these are exhaustive, but I believe that these can be done without LLM levels of compute. IMO, these are important, as the core contribution of the paper is empirical and not theoretical.

If they are all in the appendix, it would be useful to provide a summary of the key takeaway points in the main paper, with adequate references.

---------
Writing:

- Deteriorates significantly as the paper progresses.

- Also, I personally think that this method needs minimal background and the reader should not need to wait for the 5th page to see what the method is.

- Further, as much as I am a fan of figures that explain the idea, Fig 1 had marginal utility in making me understand the paper.

--------
Minor:

- Eq 9: Should the LHS be denoted as $h^*$?

----
Claims and Evidence:

- Please ignore the section below for claims and evidence (openreview does not allow me to edit those)
- Currently, I do not think the paper has enough evidence to support the claims (but I also unclear about many parts of the paper). I am happy to edit this based on authors rebuttal.

**Strengths And Weaknesses:**

Strengths:

S1: Looks at an important and understudied problem

---
Weakness:

W1: It is not clear how should the mse be estimated for OPE problems in practice. Unfortunately, that is the core building block of the entire idea.

W2: Paper is unclear in many parts

---

### Review · Reviewer_o8uf · 2024-09-05

**Recommendation:** 3
**Confidence:** 2

**Summary Of Contributions:**

The authors propose DataCOPE, a novel data-centric framework for assessing the reliability of OPE in contextual bandit settings. It estimates aleatoric and epistemic uncertainties to predict mean squared errors in OPE.

**Strengths:**

The work focuses on an important problem: given a logged dataset and a policy, how accurate are off-policy estimators? Furthermore, the angle of the work, which emphasizes the importance of the data, brings a novel and valuable perspective to the off-policy field.

I also appreciate the authors' effort in connecting OPE to LLM alignment.

**Audience:**

Yes

**Claims And Evidence:**

I don't think the claims are well supported without any explicit comparison against the existing baselines.

**Datasets And Benchmarks:**

N/A

**Extended Submissions:**

N/A

**Limitations:**

See above

**Requested Changes:**

Although I commend the authors for giving us a new perspective, I believe the work should connect more to existing problems in the OPE domains, as it is difficult to judge the practicality of this method from the current experiments.

One of the method’s motivations lies in answering whether a provided OPE dataset is appropriate to give us accurate OPE estimates in the first place. However, the experimental results show only a *correlation* between the true MSE and the estimated one, suggesting the method cannot be used for direct MSE estimation. The key characteristics, such as aleatoric and epistemic uncertainties, differ across the datasets. As the method offers only a relative comparison for different datasets/estimators, **how can we truly know whether we can proceed with OPE or we should go back to the data collection?** It is also difficult to interpret how well the method would perform in the hyper-parameter tuning and estimator selection tasks only from the correlation.

Why do authors care about answering the question, “Which dataset is most appropriate in evaluating a target policy”? I think this is not a significant problem in OPE, and simply using all the available data should be at least as good as any of its subsets.

---

I don’t think authors do justice to related work.

**Conformal methods:**  In my understanding, conformal off-policy prediction [1, 2] produces intervals for the policy value estimate. These can be used the same way as the author’s proposed epistemic and aleatoric uncertainty to evaluate how suitable the currently logged dataset is or any other questions the authors answer by their method.

**Estimator selection methods:** The authors claim [3] is the concurrent work; however, that one has been on arXiv since November 2022. In my opinion, this should be compared against. Other notable methods worth discussing are [4, 5]. Furthermore, while it is not strictly a weakness, other related works that address the same problem emerged recently that should be at least discussed in the camera-ready version if not directly compared [6, 7].


[1] Taufiq, Muhammad Faaiz, et al. “Conformal off-policy prediction in contextual bandits.” Advances in Neural Information Processing Systems 35 (2022): 31512-31524.

[2] Zhang, Yingying, Chengchun Shi, and Shikai Luo. “Conformal off-policy prediction.” International Conference on Artificial Intelligence and Statistics. PMLR, 2023.

[3] Udagawa, Takuma, et al. “Policy-adaptive estimator selection for off-policy evaluation.” Proceedings of the AAAI Conference on Artificial Intelligence. Vol. 37. No. 8. 2023.

[4] Saito, Yuta, et al. “Evaluating the robustness of off-policy evaluation.” Proceedings of the 15th ACM Conference on Recommender Systems. 2021.

[5] Su, Yi, Pavithra Srinath, and Akshay Krishnamurthy. “Adaptive estimator selection for off-policy evaluation.” International Conference on Machine Learning. PMLR, 2020.

[6] Cief, Matej, Michal Kompan, and Branislav Kveton. “Cross-Validated Off-Policy Evaluation.” arXiv preprint arXiv:2405.15332 (2024).

[7] Felicioni, Nicolò, Michael Benigni, and Maurizio Ferrari Dacrema. “AutoOPE: Automated Off-Policy Estimator Selection.” arXiv preprint arXiv:2406.18022 (2024).

**Strengths And Weaknesses:**

The method and framing of the problem are novel and interesting. However, I miss a more explicit connection to existing off-policy evaluation methods that partially study this, both with a more extensive discussion and empirical comparison. Without these, it is difficult to gauge the method's efficiency.